# Regulating algorithmic filtering on social media

**Sarah H. Cen**
MIT EECS
shcen@mit.edu

**Devavrat Shah**
MIT EECS
devavrat@mit.edu

## Abstract

By filtering the content that users see, social media platforms have the ability to influence users' perceptions and decisions, from their dining choices to their voting preferences. This influence has drawn scrutiny, with many calling for regulations on filtering algorithms, but designing and enforcing regulations remains challenging. In this work, we examine three questions. First, given a regulation, how would one design an audit to enforce it? Second, does the audit impose a performance cost on the platform? Third, how does the audit affect the content that the platform is incentivized to filter? In response to these questions, we propose a method such that, given a regulation, an auditor can test whether that regulation is met with only black-box access to the filtering algorithm. We then turn to the platform's perspective. The platform's goal is to maximize an objective function while meeting regulation. We find that there are conditions under which the regulation does not place a high performance cost on the platform and, notably, that content diversity can play a key role in aligning the interests of the platform and regulators.

## 1 Introduction

In recent years, there have been increasing calls to regulate how social media platforms *algorithmically filter* the content that appears on a user's feed. For example, one may ask that the advertisements a user sees not be based (explicitly or implicitly) on their sexual orientation [65] or that content related to public health (e.g., COVID-19) does not reflect a user's political affiliation [35].

However, translating a regulatory guideline into an auditing procedure has proven difficult. Developing such a method is the focus of this work. As our main contribution, we provide a general procedure such that, *given a regulation, an auditor can test whether the platform complies with the regulation.*

Providing a general procedure is important because, without it, auditing is destined to be reactive: auditors must design ways to enforce each regulation as they arise, and there is an inevitable delay between design and enforcement in order to test the proposed solutions.

On the other hand, designing a general procedure is challenging because audits can have unintended side effects on the social media ecosystem and its many stakeholders. As a result, we consider two additional questions in this work:

1. *How does the proposed auditing procedure affect the platform's bottom line?* We consider whether the procedure imposes a high performance cost on the platform.
2. *How does the auditing procedure affect the user's content?* We consider what type of content the platform is incentivized to show the user when the platform complies with the regulation.

Our main contributions are summarized as follows.

**Auditing procedure**. As our main contribution, we provide a procedure such that, given a regulation, an auditor can test whether the platform complies with the regulation (Section 3). We restrict our attention to regulations that can be written in *counterfactual form*, including the two examples given at the top of this Introduction. Namely, the procedure applies to any regulation that can be written as:

35th Conference on Neural Information Processing Systems (NeurIPS 2021).

"the filtering algorithm $\mathcal{F}$ should behave similarly under inputs $\mathbf{x}$ and $\mathbf{x}'$ for all $(\mathbf{x}, \mathbf{x}') \in \mathcal{S}$". How to quantify "similarity" under $\mathbf{x}$ and $\mathbf{x}'$ is crucial, and our second main contribution is to provide a precise notion of "similarity" in the context of algorithmic filtering (Section 2).

Operationally, the auditing procedure has several desirable properties. First, it needs only black-box access to the algorithm and therefore holds even if the filtering algorithm $\mathcal{F}$ changes. Second, it does not require access to the platform's users or their personal data. Third, its parameters are interpretable and easy to tune. Finally, the procedure is modular, which allows for many possible configurations.

**Provable guarantees**. To audit a counterfactual regulation, we begin by observing that algorithmic filtering is powerful (and often harmful) because *information influences decisions*: the content that a user sees can affect how they vote, whether they choose to receive a vaccine, what restaurants they frequent, and more. Therefore, if one seeks to enforce a counterfactual regulation, the notion of "similarity" that is enforced should be with respect to the outcome of interest: *the users' decisions*.

Suppose that the auditor seeks to audit content of type $T$ (e.g., the advertisements or the entire feed). Suppose the filtering algorithm $\mathcal{F}$ generates content $Z$ of type $T$ when given inputs $\mathbf{x}$, and $Z'$ when given $\mathbf{x}'$. In Theorem 1, we prove that, if $\mathcal{F}$ passes the audit, the decision-making of *any* user if shown $Z$ and their decision-making if shown $Z'$ instead are (asymptotically) indistinguishable. This guarantee—which we call *decision robustness* and formalize in Section 2—is powerful because it holds for any user and their decision-making even though the audit does not have access to the users' personal data (e.g., gender) or knowledge about the user's decision-making tendencies (e.g., how easily a user is influenced by restaurant advertisements). Providing such a strong guarantee without access to users is made possible by two well-known concepts from decision and learning theory: the hypothesis test and minimum-variance unbiased estimator. Combining these tools is one of our main technical insights and discussed in Section 4.

**Cost of regulation**. In Section 5.1, we study how the audit affects a platform's ability to maximize an objective function $R$—which we refer to as *reward*—and find that being audited does not necessarily place a high performance cost on the platform. Studying the *cost of regulation* is important because there are concerns that regulations can hurt innovation or profits, and our findings surface conditions under which a performance-regulation trade-off does not exist. Because we leave $R$ unspecified, the analysis in Section 5.1 is applicable to any $R$. As examples, $R$ could measure time spent on the platform, the number of clicks on posts, or a combination of these factors.

**Content diversity**. In Section 5.2, we turn our attention to how an audit would affect the users' content and discover an unexpected connection to content diversity. We find that, under regulation, social media platforms are *incentivized to add doses of content diversity*. Specifically, when faced with a regulation, it is in the platform's interest to ensure that the content of type $T$ that it shows users is sufficiently diverse along the dimension by which $\mathbf{x}$ and $\mathbf{x}'$ differ for all $(\mathbf{x}, \mathbf{x}') \in \mathcal{S}$. Because content diversity is not a part of the regulatory test by design, this result is unexpected and suggests that content diversity plays a key role in aligning the interests of regulators and platforms.

All proofs are given in the Appendix as well as a toy example and further discussion of the audit.

## 2 Problem statement

Consider a system with two agents: a social media platform and an auditor.

1. The **platform** selects the content that is shown to its users using a filtering algorithm $\mathcal{F} : \mathcal{X} \to \mathcal{Z}$ such that $Z = \mathcal{F}(\mathbf{x})$ is the feed produced by $\mathcal{F}$ given inputs $\mathbf{x} \in \mathcal{X}$. Here, a feed is a collection of content that is shown to a user, and $\mathbf{x}$ captures any inputs that the platform uses to filter, such as the user's interaction history, social network, available content sources, and so on. Each feed $Z = \{\mathbf{z}_1, \ldots, \mathbf{z}_m\}$ consists of $m$ pieces of content, where $\mathbf{z}_i \in \mathbb{R}^n$ for all $i \in [m]$. We assume that $\mathcal{F}$ can be written as a generative model such that $\mathcal{F}$ generates the content in $Z$ by drawing $m$ samples from a distribution $p_{\mathbf{z}}(\cdot; \boldsymbol{\theta}(\mathbf{x}))$, where $\boldsymbol{\theta}(\mathbf{x}) \in \Theta \in \mathbb{R}^r$ is unknown.[1] Note that if one is only concerned with a certain type $T$ of content (e.g., the auditor only wishes to monitor advertisements), then one can take $Z$ to be the content of type $T$ that is filtered by $\mathcal{F}$ rather than the entire feed.

---

[1] This representation is without loss of generality. For example, any deterministic mapping from $\mathbf{x}$ to $Z$ can be achieved by letting $r = nm$, $\boldsymbol{\theta}(\mathbf{x}) = (\mathbf{z}_1^\top, \ldots, \mathbf{z}_m^\top)$, and sequentially generating $\mathbf{z}_i$ from the entries of $\boldsymbol{\theta}(\mathbf{x})$.

2. The **auditor** is given a regulatory guideline that they wish to enforce. The auditor's goal is to check whether the platform's filtering algorithm $\mathcal{F}$ is in compliance with the given regulation. We assume that the auditor has black-box access to $\mathcal{F}$. In other words, the auditor can run $\mathcal{F}$ on a set of inputs $\{\mathbf{x}_j\}$ and observe its outputs $\{Z_j = \mathcal{F}(\mathbf{x}_j)\}$. Note that the inputs $\mathbf{x}$ and $\mathbf{x}'$ need not correspond to real users and could represent *hypothetical* users.

In this work, we restrict our analysis to **counterfactual regulations**. Specifically, the auditor is given a regulation in the form: "The filtering algorithm $\mathcal{F}$ should behave similarly on content of type $T$ under inputs $\mathbf{x}$ and $\mathbf{x}'$ for all $(\mathbf{x}, \mathbf{x}') \in \mathcal{S}$." In the remainder of this work, we omit references to type $T$ and refer directly to content $Z$. We give three examples of counterfactual regulations.

**Example 1.** *A regulation that prohibits targeted advertisements from being based on a user's indicated sexual orientation [65] can be written as "the advertisements shown by $\mathcal{F}$ should be similar when given two users who are identical except for their sexual orientations".*[2]

**Example 2.** *A regulation that requires articles containing medical advice on COVID-19 be robust to whether the user is left- or right-leaning can be framed as "the articles that are selected by $\mathcal{F}$ and provide medical advice on COVID-19 should be similar for left- and right-leaning users".*

**Example 3.** *A regulation prescribing that a platform not sway voting preferences beyond serving as a social network can be framed as "content about political candidates that are injected by $\mathcal{F}$ should be similar to the content a user would see from its social network without algorithmic filtering."*

The goal of this work is to enforce a regulation of the form "$\mathcal{F}$ should behave similarly under $\mathbf{x}$ and $\mathbf{x}'$ for all $(\mathbf{x}, \mathbf{x}') \in \mathcal{S}$". The question remains: *What is an appropriate notion of "similarity"?*

## 2.1 Decision robustness

We begin by observing that algorithmic filtering is powerful (and often harmful) because *information influences decisions*: the content that a user sees can affect how they vote, whether they get vaccinated, what restaurants they frequent, what items they purchase, and more. Stated differently, if algorithmic filtering did not influence users' decisions, then there would be no desire to regulate it.

Therefore, if one seeks to enforce a counterfactual regulation, the notion of "similarity" that is enforced should be with respect to the outcome of interest: the users' decisions. However, an auditor does not and should not have access to the users or their decisions (e.g., whether they get vaccinated). As such, the problem that the auditor faces can be stated as follows.

Consider a (hypothetical) user. Suppose that there are two identical worlds except that the user is shown $\mathcal{F}(\mathbf{x})$ in one world and $\mathcal{F}(\mathbf{x}')$ in the other. Because the worlds are otherwise identical, both users subsequently face an identical set of queries $\mathcal{Q}$ (e.g., where to eat, whether to get vaccinated, what to wear). Let $\mathcal{D}$ and $\mathcal{D}'$ denote the (hypothetical) decisions that the first and second users make given queries $\mathcal{Q}$. Then, the auditor enforces similarity by ensuring **decision robustness** as follows:

$\mathcal{F}$ is *decision-robust* to $(\mathbf{x}, \mathbf{x}')$ if and only if, for any user and any $\mathcal{Q}$, one cannot determine with high confidence that $\mathbf{x} \neq \mathbf{x}'$ from the decisions $\mathcal{D}$ and $\mathcal{D}'$.

Decision robustness guarantees that the decision-making behavior of any user under $\mathcal{F}(\mathbf{x})$ and $\mathcal{F}(\mathbf{x}')$ is indistinguishable with respect to $(\mathbf{x}, \mathbf{x}')$. However, ensuring decision robustness is challenging because the auditor does not have access to users or their decisions. Our objective is to *provide an auditing procedure that guarantees decision robustness given only $\mathcal{S}$ and black-box access to $\mathcal{F}$.*

## 2.2 Formalizing the auditor's goal

Recall that $\mathcal{F}$ is decision-robust to $(\mathbf{x}, \mathbf{x}')$—and therefore complies with the regulation—when, for any user and any $\mathcal{Q}$, one cannot determine with high confidence that $\mathbf{x} \neq \mathbf{x}'$ from $\mathcal{D}$ and $\mathcal{D}'$. In this section, we show that decision robustness can be expressed as a **binary hypothesis test**.

Formally, consider a pair of inputs $(\mathbf{x}, \mathbf{x}') \in \mathcal{S}$ and set of queries $\mathcal{Q}$. To "determine whether $\mathbf{x} \neq \mathbf{x}'$ from $\mathcal{D}$ and $\mathcal{D}'$" is equivalent to using $\mathcal{D}$ and $\mathcal{D}'$ to decide between the following hypotheses:

$$H_0 : \boldsymbol{\theta}(\mathbf{x}) = \boldsymbol{\theta}(\mathbf{x}') \qquad H_1 : \boldsymbol{\theta}(\mathbf{x}) \neq \boldsymbol{\theta}(\mathbf{x}') \tag{1}$$

---

[2] This example is simplified to illustrate a simple counterfactual regulation. To protect against more nuanced effects, such as proxy variables, one could modify both sexual orientation and its proxies. Producing counterfactual inputs is out of the scope of this work. We direct interested readers to other texts [58, 41, 52, 43].

To see this equivalence, observe that we can write the Markov chain $\mathbf{x} \to \boldsymbol{\theta}(\mathbf{x}) \to Z \to \mathcal{D}$. In other words, decisions $\mathcal{D}$ depend on inputs $\mathbf{x}$ only through the parameters $\boldsymbol{\theta}(\mathbf{x})$. If one cannot determine that $\boldsymbol{\theta}(\mathbf{x}) \neq \boldsymbol{\theta}(\mathbf{x}')$ from $\mathcal{D}$ and $\mathcal{D}'$, then one also cannot determine that $\mathbf{x} \neq \mathbf{x}'$.

Let $H \in \{H_0, H_1\}$ denote the true (unknown) hypothesis.[3] Let $\hat{H} \in \{H_0, H_1\}$ denote the hypothesis that is chosen, where the outcome $\hat{H} = H_1$ is equivalent to determining that $\boldsymbol{\theta}(\mathbf{x}) \neq \boldsymbol{\theta}(\mathbf{x}')$.

We say that a test $\hat{H}$ is $(1-\epsilon)$-**confident** that $\boldsymbol{\theta}(\mathbf{x}) \neq \boldsymbol{\theta}(\mathbf{x}')$ if $\hat{H} = H_1$ and $\mathbb{P}(\hat{H} = H_1 | H = H_0) \leq \epsilon \in [0, 1]$.[4] While one would like the test to be $(1 - \epsilon)$-confident, a trivial test that always chooses $H_0$ is $(1 - \epsilon)$-confident but has 100 percent error when $H = H_1$. Therefore, one would also like the test to satisfy $\mathbb{P}(\hat{H} = H_0 | H = H_1) \leq \alpha$ for some small $\alpha \in [0, 1]$, but not all $\alpha \in [0, 1]$ may be achievable while retaining the property of $(1 - \epsilon)$-confidence.

To this end, we turn to the **uniformly most powerful unbiased (UMPU)** test [21, 47], which one can think of as follows: if the UMPU test cannot confidently determine that $\boldsymbol{\theta}(\mathbf{x}) \neq \boldsymbol{\theta}(\mathbf{x}')$, then no other reasonable test can. Formally, suppose that one would like to find a test that maximizes the true positive rate (TPR) while ensuring the false positive rate (FPR) is at most $\epsilon$ such that the test solves:

$$\max_{\hat{H}} \mathbb{P}(\hat{H} = H_1 | H = H_1) \qquad \text{s.t.} \qquad \mathbb{P}(\hat{H} = H_1 | H = H_0) \leq \epsilon, \tag{2}$$

If a test $\hat{H}$ solves (2) for all $\boldsymbol{\theta}(\mathbf{x}), \boldsymbol{\theta}(\mathbf{x}') \in \Theta$, then it is the **uniformly most powerful (UMP)** test. The UMPU test is the UMP test among all unbiased tests, where a test $\hat{H}$ is **unbiased** if $\mathbb{P}(\hat{H} = H_1 | H = H_1) \geq \delta \geq \mathbb{P}(\hat{H} = H_1 | H = H_0)$ for some $\delta \in [0, 1]$.[5]

Let $\hat{H}_\epsilon^*$ denote the UMPU test, if it exists, and $\alpha^*(\epsilon) = \mathbb{P}(\hat{H}_\epsilon^* = H_0 | H = H_1)$. Intuitively, $\hat{H}_\epsilon^*$ is the test that is best at detecting when $\mathcal{F}$ is not decision-robust (i.e., it maximizes the TPR) while making sure that it rarely falsely accuses $\mathcal{F}$ of not being decision-robust (i.e., its FPR is at most $\epsilon$) among all reliable (i.e., unbiased) tests. Given the UMPU test, decision-robustness can be formalized as follows. For $\epsilon, \alpha \in [0, 1]$ and when the UMPU test exists,

$$\mathcal{F} \text{ is } (\epsilon, \alpha)\text{-}decision\text{-}robust \text{ to } (\mathbf{x}, \mathbf{x}') \iff \text{ for any } Q, \mathbb{P}(\hat{H}_\epsilon^* = H_0 | H = H_1) \leq \alpha.$$

**The goal of the auditing procedure**. Therefore, determining whether a platform's filtering algorithm $\mathcal{F}$ complies with a counterfactual regulation comes down to determining whether, for all $Q$ and $(\mathbf{x}, \mathbf{x}') \in \mathcal{S}$, the UMPU test cannot confidently reject $H_0$ given $\mathcal{D}$ and $\mathcal{D}'$. However, this task is not straightforward because the auditor's goal is to provide a guarantee on how $\mathcal{F}$ affects users' decisions *without* access to the users or their decisions (i.e., without $Q$, $\mathcal{D}$, or $\mathcal{D}'$). In this work, we show that it is possible to guarantee approximate asymptotic decision-robustness *given only $\mathcal{S}$ and black-box access to $\mathcal{F}$* using insights from statistical learning and decision theory.

# 3 Auditing procedure

In this section, we present a procedure such that, *given a regulation on algorithmic filtering that is expressed in counterfactual form, an auditor can test whether the platform's filtering algorithm is in compliance with the regulation*. In Section 4, we show that, if $\mathcal{F}$ passes the audit, then $\mathcal{F}$ is approximately asymptotically decision-robust. In Section 5, we study the cost of regulation and find that there are conditions under which the audit does not place a performance cost on the platform.

## 3.1 Notation and definitions

Before proceeding, we require some notation and definitions. Recall that $\mathcal{F}$ generates $Z$ by drawing $m$ samples from $p_{\mathbf{z}}(\cdot; \boldsymbol{\theta}(\mathbf{x}))$, where $\boldsymbol{\theta}(\mathbf{x})$ is unknown. In statistical inference [46], an *estimator* is a mapping $\mathcal{L} : \mathcal{Z} \to \Theta$ such that $\mathcal{L}(Z)$ is an estimate of the parameters $\boldsymbol{\theta}$ that generated $Z$.

**Definition 1.** *An estimator $\mathcal{L} : \mathcal{Z} \to \Theta$ is* unbiased *if and only if $\mathbb{E}_{p_{\mathbf{z}}(\cdot; \boldsymbol{\theta})}[\mathcal{L}(Z)] = \boldsymbol{\theta}$ for all $\boldsymbol{\theta} \in \Theta$.*

---

[3] Although the auditor has access to $S$ and knows whether $\mathbf{x} \neq \mathbf{x}'$, decision-robustness requires that one cannot determine this fact from $\mathcal{D}$ and $\mathcal{D}'$. Therefore, the hypothesis test treats $\mathbf{x}$ and $\mathbf{x}'$ as unknown.

[4] $\mathbb{P}$ is taken with respect to $p_{\mathbf{z}}(\cdot; \boldsymbol{\theta}(\mathbf{x}))$ and $p_{\mathbf{z}}(\cdot; \boldsymbol{\theta}(\mathbf{x}'))$.

[5] Intuitively, an unbiased test $\hat{H}$ ensures that the probability that $\hat{H}$ chooses $\mathbf{x} \neq \mathbf{x}'$ is always higher when $H_1 : \mathbf{x} \neq \mathbf{x}$ is true than when $H_0 : \mathbf{x} = \mathbf{x}'$ is true.

---

**Algorithm 1:** Modular version of auditing procedure

---

**Input:** Regulation parameter $\epsilon \in [0, 1]$; model family $\Theta \subset \mathbb{R}^r$; black-box access to the filtering algorithm $\mathcal{F} : \mathcal{X} \to \mathcal{Z}$; a pair of counterfactual inputs $(\mathbf{x}, \mathbf{x}') \in \mathcal{S}$.

**Result:** $\hat{H}_\epsilon = H_0$ if the test is passed; $\hat{H}_\epsilon = H_1$, otherwise.

**1** $\tilde{\boldsymbol{\theta}} \leftarrow \mathcal{L}^+(\mathcal{F}(\mathbf{x}))$;

**2** $\tilde{\boldsymbol{\theta}}' \leftarrow \mathcal{L}^+(\mathcal{F}(\mathbf{x}'))$;

**3** **if** $(\tilde{\boldsymbol{\theta}} - \tilde{\boldsymbol{\theta}}')^\top I(\tilde{\boldsymbol{\theta}})(\tilde{\boldsymbol{\theta}} - \tilde{\boldsymbol{\theta}}') \geq \frac{2}{m}\chi_r^2(1 - \epsilon)$ **then**

**4** $\quad$ Return $\hat{H}_\epsilon = H_1$;

**5** **end**

**6** Return $\hat{H}_\epsilon = H_0$;

---

**Definition 2.** *When it exists, the* minimum-variance unbiased estimator (MVUE) $\mathcal{L}^+ : \mathcal{Z} \to \Theta$ *is an estimator that is unbiased and has the lowest variance among all unbiased estimators, i.e., satisfies* $\mathbb{E}_{p_\mathbf{z}(\cdot\,;\boldsymbol{\theta})}[(\mathcal{L}^+(Z) - \boldsymbol{\theta})^2] \leq \mathbb{E}_{p_\mathbf{z}(\cdot\,;\boldsymbol{\theta})}[(\mathcal{L}(Z) - \boldsymbol{\theta})^2]$ *for all unbiased* $\mathcal{L} : \mathcal{Z} \to \Theta$ *and all* $\boldsymbol{\theta} \in \Theta$.

Let $\chi_r^2$ denote the chi-squared distribution with $r$ degrees of freedom and $\chi_r^2(q)$ be defined such that $\mathbb{P}(v \leq \chi_r^2(q)) = q$ where $v \sim \chi_r^2$. Lastly, let $I(\boldsymbol{\theta}) \in \mathbb{R}^{r \times r}$ denote the Fisher information matrix at $\boldsymbol{\theta}$. An exact definition of $I(\boldsymbol{\theta})$ is given in the Appendix. Intuitively, $I(\boldsymbol{\theta})$ captures how well an estimator can learn $\boldsymbol{\theta}$ from $Z$. As a simple example, suppose $\mathbf{z}_i$ are drawn i.i.d. from $\mathcal{N}(\mu, \sigma^2)$, where $\sigma^2$ is known, $r = 1$, and $\theta = \mu$. It takes more samples to accurately estimate $\mu$ when the variance $\sigma^2$ is large, and, as expected, the Fisher information scales with $1/\sigma^2$.

### 3.2 The audit

In this section, we present the auditing procedure. Recall that a counterfactual regulation requires that $\mathcal{F}$ behave similarly under $\mathbf{x}$ and $\mathbf{x}'$ for all $(\mathbf{x}, \mathbf{x}') \in \mathcal{S}$. Algorithm 1 provides a test for determining whether $\mathcal{F}$ complies with the given regulation for a pair of inputs $(\mathbf{x}, \mathbf{x}') \in \mathcal{S}$. To test other pairs in $\mathcal{S}$, simply repeat Algorithm 1 and modify $\mathbf{x}$ and $\mathbf{x}'$ accordingly. If $\hat{H}_\epsilon = H_1$ for any pair, then the platform does not pass the audit. Below, we list and explain several characteristics of the audit.

**Modularity**. Algorithm 1 is designed to be modular. Modularity allows the auditor to construct the audit as they wish. For example, the auditor may wish to add more pairs to $\mathcal{S}$ or to repeat the audit every month. Alternatively, the auditor may require not that $\mathcal{F}$ behave similarly under $\mathbf{x}$ and $\mathbf{x}'$ for *all* $(\mathbf{x}, \mathbf{x}') \in \mathcal{S}$ but for at least $(1 - \alpha) \in [0, 1]$ of them.[6] To do so, the auditor can run Algorithm 1 over $\mathcal{S}$ and, if the number of times it returns $H_1$ exceeds $\alpha|\mathcal{S}|$, then the platform does not pass the audit. Modularity also allows the auditor to see the pairs $(\mathbf{x}, \mathbf{x}')$ for which $\mathcal{F}$ fails the test.

**Tunable parameter**. One benefit of the procedure is that the tunable parameter $\epsilon$ has an intuitive meaning. We see in Section 4.1 that $\epsilon$ is a maximum FPR. Capping the FPR ensures that the auditor is not distracted by red herrings and prevents the auditor from investing the resources needed to investigate (or bring a case against) the platform unless they are at least $(1 - \epsilon)$-confident that the platform violates the regulation. Decreasing $\epsilon \in [0, 1]$ reduces the number of false positives while increasing $\epsilon$ makes the regulation more strict (at the risk of receiving more false positives).

**Other advantages**. In addition to the benefits regarding modularity and $\epsilon$ discussed above, this procedure has two additional advantages. First, the procedure does not require access to users or their personal data. Second, it requires only black-box access to $\mathcal{F}$, which means that an auditor does not need to know the inner-workings of $\mathcal{F}$ (there is often resistance to giving auditors full access to $\mathcal{F}$) and, perhaps more importantly, the procedure works even when $\mathcal{F}$ changes internally.

**When the MVUE does not exist, use the MLE**. Recall that $\mathcal{L}^+$ denotes the MVUE. We will see that there is a theoretical justification for using the MVUE (Proposition 2). However, there are cases in which the MVUE does not exist but the maximum likelihood estimator (MLE) does. The MLE is a good substitute for the MVUE because the MVUE and MLE are often asymptotically equivalent [59]. When this asymptotic equivalence holds, the main theoretical guarantee—namely, Theorem 1—holds when $\mathcal{L}^+$ is taken to be the MLE instead of the MVUE.

---

[6]Here, $\alpha \in [0, 1]$ would correspond to the maximum allowable false negative rate (FNR).

**Symmetry**. Algorithm 1 is not symmetric with respect to $\boldsymbol{\theta}$ and $\boldsymbol{\theta}'$ (or, equivalently, $\mathbf{x}$ and $\mathbf{x}'$). This can be useful if the auditor would like to have a baseline input $\mathbf{x}$ and run Algorithm 1 over different $\mathbf{x}'$. If the auditor would like symmetry, they may wish to run Algorithm 1 twice, swapping the order of $\mathbf{x}$ and $\mathbf{x}'$, or to alter the Fisher information matrix in Line 3 to be $I((\tilde{\boldsymbol{\theta}} + \tilde{\boldsymbol{\theta}}')/2)$, if it exists.

**Choice of** $\Theta$. Recall that $\Theta$ captures the set of possible generative models. In choosing the model family $\Theta$, the auditor may find that a simple $\Theta$ is more tractable and interpretable while a complex $\Theta$ is more general. As explained in Section 4.2, $\Theta$ can also be viewed as the set of possible cognitive models that users employ when making decisions. Therefore, the auditor may wish to choose $\Theta$ to be just rich enough to mirror the complexity of common cognitive models.

## 4 Explaining the procedure and its theoretical guarantees

Recall from Section 2 that a filtering algorithm $\mathcal{F}$ complies with a counterfactual regulation if $\mathcal{F}$ is decision-robust. In Section 4.1, we show that, if the platform passes the audit in Algorithm 1, then $\mathcal{F}$ is guaranteed to be approximately asymptotically decision-robust. In Section 4.2, we provide insights on the role of the MVUE. All proofs are given in the Appendix.

### 4.1 Guarantee on the audit's effectiveness

**Theorem 1.** *Consider* (1). *Let* $\boldsymbol{\theta}^* = (\boldsymbol{\theta}(\mathbf{x}) + \boldsymbol{\theta}(\mathbf{x}'))/2$. *Suppose that* $\mathbf{z}_i$ *and* $\mathbf{z}'_i$ *are drawn i.i.d. from* $p(\cdot; \boldsymbol{\theta}(\mathbf{x}))$ *and* $p(\cdot; \boldsymbol{\theta}(\mathbf{x}'))$, *respectively, for all* $i \in [m]$ *and* $\mathcal{P} = \{p_{\mathbf{z}}(\cdot; \boldsymbol{\theta}) : \boldsymbol{\theta} \in \Theta\}$ *is a regular exponential family that meets the regularity conditions stated in Appendix B. If* $\hat{H}$ *is defined as:*

$$\hat{H} = H_1 \iff (\mathcal{L}^+(Z) - \mathcal{L}^+(Z'))^\top I(\boldsymbol{\theta}^*)(\mathcal{L}^+(Z) - \mathcal{L}^+(Z')) \geq \frac{2}{m}\chi_r^2(1-\epsilon), \qquad (3)$$

*then* $\mathbb{P}(\hat{H} = H_1 | H = H_0) \leq \epsilon$ *as* $m \to \infty$. *If* $r = 1$, *then* $\lim_{m\to\infty} \mathbb{P}(\hat{H} = H_0 | H = H_1) = \alpha^*(\epsilon)$.

**Understanding the result**. Recall that the goal of an auditor is to determine whether the platform's filtering algorithm $\mathcal{F}$ is compliant with a given regulation by determining whether $\mathcal{F}$ is decision-robust. Theorem 1 confirms that the audit in Algorithm 1 enforces approximate asymptotic decision robustness. To see this connection, observe that the test in (3) is identical to the test in Algorithm 1 with one substitution—$\mathcal{L}^+(Z)$ is replaced by $\boldsymbol{\theta}^*$—which implies that the test in (3) is asymptotically equivalent to the audit. Therefore, Theorem 1 establishes that, if Algorithm 1 returns $\hat{H}_\epsilon = H_1$, then the auditor is $(1 - \epsilon)$-confident that $\mathcal{F}$ is not decision-robust as $m \to \infty$.[7]

Intuitively, if the platform passes the audit in Algorithm 1, then the decision-making of any user that is shown $\mathcal{F}(\mathbf{x})$ is guaranteed to be $\epsilon$-indistinguishable from their decision-making should they have been shown $\mathcal{F}(\mathbf{x}')$ instead. Importantly, this guarantee is provided *without* access to users or their decisions with the help of insights from statistical learning theory (see proof of Theorem 1).

**A few remarks**. First, $\epsilon$ is a false positive rate (FPR). Decreasing $\epsilon$ increases the confidence that the auditor would like to have should it pursue action against the platform (see Section 3). Second, recall from Section 3 that Algorithm 1 is modular and that, in most cases, one may wish to repeat it several times with different inputs. In that case, the result of Theorem 1 holds for each individual run. Lastly, Theorem 1 provides an asymptotic guarantee on the audit. In the next section, we show that, by using the MVUE, Algorithm 1 ensures a notion of decision robustness even for finite $m$.

### 4.2 Insight on the MVUE

Recall that the auditor's goal is to provide a guarantee with respect to users' decisions, but the auditor *does not have access to the users or their decisions*. In this section, we provide intuition for how the auditor enforces decision robustness without this information, and we explain how the use of the MVUE in Algorithm 1 allows the auditor to enforce a notion of decision robustness for finite $m$. One can think of the MVUE as providing an "upper bound" on how much content $Z$ can influence a user's decisions. This result is useful because it allows the auditor to reason about users' decisions without access to users or their decisions, both which can be expensive or unethical to obtain.

---

[7]We say that, if $\mathcal{F}$ passes the audit, it is *approximately* decision-robust because $\hat{H}_\epsilon$ is not the UMPU test, as defined in Section 2. Obtaining a UMPU test is generally difficult for $r > 1$, but the test $\hat{H}_\epsilon$ is not far from the UMPU test, as demonstrated by the fact that it is the UMPU test when $r = 1$.

**User model**. A user's decision-making process proceeds in three steps: the user observes information, updates their internal belief based on this information, then uses their belief to make decisions. Let $p_{\mathbf{z}}(\cdot; \hat{\boldsymbol{\theta}})$ denote the belief of a (hypothetical) user who is shown content $Z$, where $\hat{\boldsymbol{\theta}} \in \Theta$.[8] Let the influence that $Z$ has on a user's belief be denoted by $\mathcal{L} : \mathcal{Z} \to \Theta$ such that $\hat{\boldsymbol{\theta}} = \mathcal{L}(Z)$.[9]

**Example 4.** *As a highly simplified example, suppose $\Theta = [-1, 1] \times [0, \infty)$ and $p_{\mathbf{z}}(\cdot; \hat{\boldsymbol{\theta}})$ is a Gaussian distribution with mean $\hat{\theta}_1$ and variance $\hat{\theta}_2$, where $\hat{\theta}_1 = -1$ implies that the user does not believe that vaccines are effective, $\hat{\theta}_1 = 1$ implies the opposite, and $\hat{\theta}_2$ scales with the user's uncertainty in their belief. If a user is easily influenced, then they might develop the belief $\mathcal{L}(Z) = (-0.8, 0.1)$ after being shown content $Z$ with anti-vaccine content. Alternatively, the user could be confidently pro-vaccine and very stubborn so that no matter what content they see, $\mathcal{L}(\cdot) = (1, 0)$.*

Suppose the user is given a query $Q \in \mathcal{Q}$, for which the user decides between two options: $A_0$ and $A_1$, e.g., whether or not to get vaccinated. (Note that any decision between a finite number of options can be written as a series of binary decisions.) Internally, the user places a value on each choice such that, *if the user were given $\boldsymbol{\theta}$*, the user would choose $A_0$ if $v_0(\boldsymbol{\theta}) \geq v_1(\boldsymbol{\theta})$ and $A_1$, otherwise.

However, $\boldsymbol{\theta}$ is unknown. Instead, the user has a belief $\hat{\boldsymbol{\theta}}$ that they use to infer whether $v_0(\boldsymbol{\theta}) \geq v_1(\boldsymbol{\theta})$. For example, a user does not know the ground truth values of being vaccinated versus unvaccinated, which may depend on many factors, such as the user's underlying medical conditions. In the absence of these values, the user forms a belief about the value of receiving a vaccine based on information that they ingest. The user's decision-making process is therefore a hypothesis test between:

$$G_0 : v_0(\boldsymbol{\theta}) \geq v_1(\boldsymbol{\theta}) \qquad G_1 : v_1(\boldsymbol{\theta}) > v_0(\boldsymbol{\theta}). \tag{4}$$

The following result motivates the use of the MVUE $\mathcal{L}^+$ in the audit by demonstrating that the MVUE enforces a finite-sample version of decision robustness.

**Proposition 2.** *Consider (4). Let $G \in \{G_0, G_1\}$ denote the true (unknown) hypothesis. Suppose that $v_0, v_1 : \Theta \to \mathbb{R}$ are affine mappings and there exists $\mathbf{u} : \mathbb{R}^n \to \mathcal{U}$ such that, for $\mathcal{F}(\mathbf{x}) = \{\mathbf{z}_1, \ldots, \mathbf{z}_m\}$, one can write $\mathbf{u}(\mathbf{z}_i) \sim p_{\mathbf{u}}(\cdot; v_1(\boldsymbol{\theta}(\mathbf{x})) - v_0(\boldsymbol{\theta}(\mathbf{x})))$ for all $i \in [m]$. Then, if the UMP (or UMPU) test with a maximum FPR of $\rho$ exists, it is given by the following decision rule: reject $G_0$ (choose $A_1$) when the minimum-variance unbiased estimate $\tilde{\boldsymbol{\theta}} = \mathcal{L}^+(Z)$ satisfies $v_1(\tilde{\boldsymbol{\theta}}) - v_0(\tilde{\boldsymbol{\theta}}) > \eta_\rho$ where $\mathbb{P}(v_1(\tilde{\boldsymbol{\theta}}) - v_0(\tilde{\boldsymbol{\theta}}) > \eta_\rho | G = G_0) = \rho$; otherwise, accept $G_0$ (choose $A_0$).*

**Interpretation and implications**. The auditor is interested in how users react to their content, which is captured by $\mathcal{L}$. However, $\mathcal{L}$ may difficult or even unethical to obtain. For example, an auditor may wish to infer how advertisements affects a user's behavior, but doing so may require access to the user's personal data. Proposition 2 says that, under the stated conditions, if one wishes to study the impact of content on users' decisions, one can focus on the MVUE $\mathcal{L}^+$ because, *among all possible users, the one whose decisions are most influenced by their content is the hypothetical user given by the $\mathcal{L}^+$*. One can think of this hypothetical user as the "most gullible user". Because Line 3 in Algorithm 1 requires $\mathcal{L}^+(Z)$ and $\mathcal{L}^+(Z')$ are sufficiently close, the audit enforces approximate decision robustness by ensuring that the counterfactual beliefs $\mathcal{L}^+(Z)$ and $\mathcal{L}^+(Z')$—and therefore the counterfactual decisions $\mathcal{D}$ and $\mathcal{D}'$—of the most gullible user are indistinguishable.

**Understanding the MVUE**. For intuition on why the MVUE is the "most gullible user", recall that the MVUE is the unbiased estimator with the lowest variance. Suppose that a user's estimate $\mathcal{L}(Z)$ differs from $\mathcal{L}^+(Z)$. By definition, this estimate is biased or has higher variance. When biased, the user's estimate is consistently pulled by some factor other than $Z$. For example, a user who remains pro-vaccine no matter what content they see has a biased estimator. When the user's estimate has higher variance than the MVUE, it is an indication that the user places less confidence than the MVUE in what they glean from $Z$. For example, the user could be skeptical of what they see on social media or scrolling very quickly and only reading headlines. In this way, the MVUE corresponds to the user who "hangs on every word"—whose decisions are most affected by their content $Z$.

---

[8]The use of distributions to represent beliefs is common in the cognitive sciences [22]. Note that, although representing beliefs as distributions is borrowed from Bayesian inference [10], our representation does *not* require that people are Bayesian (update their beliefs according to Bayes' rule), which is contested [31].

[9]A user's belief may be impacted by information other than $Z$ (e.g., the user's previous belief, news that they receive from friends, or content that they view on other platforms). This can be modeled by letting $\mathcal{L} : \mathcal{Z} \times \mathcal{J} \to \Theta$ where $J \in \mathcal{J}$ captures off-platform information. Because it does not change our results (see the Appendix), we use $\mathcal{L} : \mathcal{Z} \to \Theta$ for notational simplicity.

# 5 Cost of regulation and the role of content diversity

In this section, we turn our attention to how the auditing procedure affects (a) the platform's ability to maximize an objective function $R$ and (b) the type of content the platform is incentivized to filter when compliant with a regulation. In Section 5.1, we find that there are conditions under which the audit does not place a performance cost on the platform and, intuitively, this occurs when the platform has enough degrees of freedom with which to filter. We show in Section 5.2 that one of the ways the platform can increase $R$ while complying with the regulation is to incorporate sufficient content diversity. Because diversity does not appear in the audit by design, this result suggests that content diversity can align the interests of regulators and platforms. All proofs are given in the Appendix.

## 5.1 Cost of regulation

Suppose that the platform's goal is to maximize an objective function $R : \mathcal{Z} \times \mathcal{X} \to \mathbb{R}$—which we call *reward*—while passing the audit. For example, $R$ could be a measure of user engagement, user satisfaction, content novelty, or a combination of these and other factors. We leave $R$ unspecified, which means that our analysis holds for any choice of $R$, unless otherwise stated.[10]

Recall that $\mathcal{Z}$ denotes the set of all possible feeds (or collections of type-$T$ content). Complying with a counterfactual regulation defined by $\mathcal{S}$ is equivalent to restricting the platform's choice of content to a subset $\mathcal{Z}(\mathcal{S}) \subset \mathcal{Z}$, which we call the **feasible set** under $\mathcal{S}$. Intuitively, the stricter the regulation, the smaller the feasible set $\mathcal{Z}(\mathcal{S})$. If there is no regulation, then $\mathcal{Z}(\mathcal{S}) = \mathcal{Z}$. As such, the platform's goal to maximize $R$ given inputs $\mathbf{x}$ while complying with the regulation can be expressed as:

$$Z \in \arg \max_{W \in \mathcal{Z}(\mathcal{S})} R(W, \mathbf{x}).$$

Platforms are often interested in whether a regulation imposes a performance cost. To make this notion precise, we define the **cost of regulation** as follows.

**Definition 3.** *The* cost of regulation *for inputs* $\mathbf{x}$ *is:* $C = \max_{W \in \mathcal{Z}} R(W, \mathbf{x}) - \max_{W \in \mathcal{Z}(\mathcal{S})} R(W, \mathbf{x})$.

A low cost of regulation implies that the platform can meet regulation without sacrificing much reward, while a high cost of regulation implies that there is a strong performance-regulation trade-off.

**Performance-regulation trade-off**. Suppose that the feasible set shrinks from $\mathcal{Z}$ to $\mathcal{Z}(\mathcal{S})$. Then, the maximum achievable reward is affected in one of two ways. If the feasible set shrinks such that all reward-maximizing solutions in $\mathcal{Z}$ are not contained in $\mathcal{Z}(\mathcal{S})$, then the maximum reward *decreases*, and the cost of regulation *increases*. Alternatively, if the feasible set shrinks but at least one reward-maximizing solution in $\mathcal{Z}$ is contained in $\mathcal{Z}(\mathcal{S})$, then the maximum reward stays the same, and the cost of regulation *does not increase*. Therefore, as long as $\mathcal{Z}(\mathcal{S})$ preserves at least one (near) optimal solution, the cost of regulation is low. The following result formalizes this notion. For this result, we overload the notation $R$ such that $R(\mathcal{L}^+(Z), \mathbf{x})$ denotes $R(Z, \mathbf{x})$.

**Theorem 3.** *Suppose there exists $\Omega \subset [r]$ where $1 < |\Omega| < r$ such that $R(\boldsymbol{\theta}_1, \mathbf{x}) = R(\boldsymbol{\theta}_2, \mathbf{x})$ if $\theta_{1,i} = \theta_{2,i}$ for all $i \notin \Omega$. Suppose that, for any $\boldsymbol{\theta} \in \Theta$, $\beta > 0$ and $\mathbf{v} \in \mathbb{R}^r$, there exist a vector $\bar{\boldsymbol{\theta}}_\Omega$ where $\bar{\theta}_{\Omega,i} = 0$ for all $i \notin \Omega$ and a constant $\kappa > 0$ such that $\mathbf{v}^\top I(\boldsymbol{\theta} + \kappa \bar{\boldsymbol{\theta}}_\Omega)\mathbf{v} < \beta$ and $\boldsymbol{\theta} + \kappa \bar{\boldsymbol{\theta}}_\Omega \in \Theta$. Then, if $m < \infty$, there exists a set $\mathcal{Z}$ such that the cost of regulation for $\mathbf{x}$ under Algorithm 1 is $0$.*

**Interpretation of the result**. Imposing a regulation restricts the platform's feasible set, which may place a performance cost on the platform. However, a high cost of regulation is not inevitable. Indeed, if the feasible set contains at least one (near) optimal solution, then the cost of regulation is low. Theorem 3 provides a set of conditions under which the cost of regulation is low. Intuitively, the result states that, when $R$ is independent of at least one element in the parameter vector $\boldsymbol{\theta}$ and that element has sufficient leverage over the Fisher information, then as long as the amount of content in $Z$ is finite (i.e., $m < \infty$) and the available content $\mathcal{Z}$ is expressive enough, then the platform can always construct a $Z$ from $\mathcal{Z}$ that passes the audit without sacrificing reward.

One may ask whether the conditions in Theorem 3 are feasible. To illustrate that the conditions are achievable, consider the following highly simplified example.

---

[10]There are settings in which $R$ is time-varying, e.g., when a platform's sources of revenue change with time. Making $R$ time-varying does not change our analysis or findings. Therefore, we leave $R$ static for simplicity.

**Example 5.** *Suppose $\boldsymbol{\theta} = (\mu, \sigma^2)$, $\Theta = \mathbb{R} \times \mathbb{R}_{\geq 0}$, and $\mathcal{P} = \{\mathcal{N}(\mu, \sigma^2) : (\mu, \sigma^2) \in \Theta\}$. In other words, we consider the family of 1-D Gaussian distributions. If $R$ is a function of the mean $\mu$ but not the variance $\sigma^2$, then Theorem 3 applies. To see this, observe that $\Omega = \{2\}$ and let $\bar{\boldsymbol{\theta}}_\Omega = (0, 1)$. Recalling that $I((\mu, \sigma^2)) = \text{diag}(\sigma^{-2}, \sigma^{-4}/2)$, the entries of $I(\boldsymbol{\theta})$ can be made arbitrarily small by increasing $\theta_2 = \sigma^2$. By Line 3 in Algorithm 1, the smaller the entries of $I(\boldsymbol{\theta})$, the easier it is for the platform to pass the audit. Therefore, if $Z^*$ is high-reward but not in the feasible set, the platform can still achieve $R(Z^*, \mathbf{x})$ by increasing the content's variance to obtain a new collection of content $Z$ that is in the feasible set. As long as $Z$ and $Z^*$ share $\mu$, $R(Z, \mathbf{x}) = R(Z^*, \mathbf{x})$.*

Theorem 3 provides conditions under which there is *no cost of regulation*. These conditions can be relaxed if we are interested in scenarios for which the cost of regulation is low but not zero. We build this intuition in the next section, studying one of the ways a platform can achieve high reward while remaining compliant with regulation.

## 5.2 Content diversity

In this section, we show that one way that the platform can increases its reward while complying with the regulation is to ensure that the filtered content is sufficiently diverse. These results suggest that *content diversity may help to align the interests of regulators and platforms.*

We first formalize content diversity, then show how it relates to passing the proposed audit.

**Definition 4.** *For $\mathbf{v} \in \mathbb{R}^r$ and $Z_0, Z_1 \in \mathcal{Z}$, content $Z_0$ is more* diverse *than $Z_1$ along $\mathbf{v}$ if the Fisher information matrices at $\mathcal{L}^+(Z_0)$ and $\mathcal{L}^+(Z_1)$ satisfy: $\mathbf{v}^\top(I(\mathcal{L}^+(Z_1)) - I(\mathcal{L}^+(Z_0)))\mathbf{v} > 0$.*

**Interpretation**. This definition says that the "smaller" the Fisher information, the higher the content diversity. The Fisher information matrix $I(\boldsymbol{\theta})$ can be viewed as a measure of how easy it is to learn $\boldsymbol{\theta}$ from the content $Z$ that is generated by $p_{\mathbf{z}}(\cdot; \boldsymbol{\theta})$. Consequently, when $Z$ and $Z'$ have low content diversity along $(\boldsymbol{\theta}(\mathbf{x}) - \boldsymbol{\theta}(\mathbf{x}'))$, an auditor can, without much effort, learn that $\boldsymbol{\theta}(\mathbf{x})$ is different from $\boldsymbol{\theta}(\mathbf{x}')$ and therefore that $\mathbf{x} \neq \mathbf{x}'$. Recall from Section 2.1 that being able to say with high confidence that $\mathbf{x} \neq \mathbf{x}'$ implies that $\mathcal{F}$ is not decision-robust and therefore does not comply with regulation. In this way, low content diversity reduces the likelihood that $\mathcal{F}$ passes the audit.[11] Note that the Fisher information matrix captures two notions of diversity—the diversity of topics in $Z$ and the diversity of perspectives on each given topic in $Z$—simultaneously.

**Connection between content diversity and the cost of regulation**. Recall from Algorithm 1 that $\mathcal{F}$ passes the audit when $(\tilde{\boldsymbol{\theta}} - \tilde{\boldsymbol{\theta}}')^\top I(\tilde{\boldsymbol{\theta}})(\tilde{\boldsymbol{\theta}} - \tilde{\boldsymbol{\theta}}')$ is below some threshold. The platform can therefore pass the audit by ensuring that $I(\tilde{\boldsymbol{\theta}})$ is sufficiently "small". By Definition 4, whether the Fisher information is "small" is precisely an indication of content diversity.

In this way, increasing content diversity gives the platform more leeway when filtering. By "shrinking" the Fisher information, the platform obtains more flexibility in setting $(\tilde{\boldsymbol{\theta}} - \tilde{\boldsymbol{\theta}}')$. Stated differently, if the Fisher information $I(\tilde{\boldsymbol{\theta}})$ is "large", then the platform is more constrained because $(\tilde{\boldsymbol{\theta}} - \tilde{\boldsymbol{\theta}}')$ must be very small in order for the platform to pass the audit. Therefore, if the platform has a high-reward $Z$ that does not pass the audit (i.e., is not in the feasible set), then the platform can generally maintain a high reward while complying with the regulation by adding content diversity.[12]

Because content diversity is not part of the audit by design, this result is unexpected. It states that the audit naturally incentivizes the platform to include a sufficient amount of content diversity with respect to $\mathbf{x} - \mathbf{x}'$. Returning to Example 1, if regulators require that medical advice on COVID-19 be robust to whether a user is left- or right-leaning, then the differences between the medical advice shown to users across the political spectrum is captured by $\mathbf{x} - \mathbf{x}'$. Adding content diversity along this dimension means that right-leaning users receive medical advice on COVID-19 not only from right-leaning news outlets, but also from left-leaning ones, and vice versa.

---

[11]Content diversity can also be understood in the context of Section 4.2. Suppose that the content diversities of $Z$ and $Z'$ are very low. For example, suppose that $Z$ contains only pro-vaccine content and $Z'$ contains only anti-vaccine content. Then, the MVUE would learn a strong relationship between vaccines with positive outcomes from $Z$ and vice versa for $Z'$. If $\mathcal{Q}$ contains a query about vaccines, $\mathcal{D}$ and $\mathcal{D}'$ would reflect these strong beliefs. In this way, $\mathcal{F}$ is less likely to be decision-robust when the content diversity is low. On the other hand, if the content for all users contains both pro- and anti-vaccine content, then $\mathcal{D}$ and $\mathcal{D}'$ are more similar.

[12]The platform does not increase content diversity indefinitely because the platform must also ensure that other terms in Line 3—specifically, $(\tilde{\boldsymbol{\theta}} - \tilde{\boldsymbol{\theta}}')$—do not cause the platform to fail the audit.

# 6 Background & related work

Algorithmic filtering [28, 15] has the potential to greatly improve the user experience, but it can also yield unwanted side effects, like the spread of fake news [49, 23, 25], over-representation of polarizing opinions due to comment ranking [61], amplification of echo chambers due to filter bubbles [34, 56], or advertising of products based on discriminatory judgments about user interests [62, 63, 44]. Although the severity of these effects is contested—for example, some studies argue that political polarization and echo chambers are not always products of internet use or data-driven algorithms [14, 40]—social media platforms are under rising scrutiny.

In response, some platforms have begun to self-regulate [51, 2]. For example, Facebook has established an internal "Supreme Court" that reviews the company's decisions [55], Twitter has banned accounts associated with the spread of conspiracy theories [8], YouTube has removed videos it views as encouraging violence [64], and so on. Self-regulatory practices offer various benefits, such as the ability to adapt quickly to a changing social media ecosystem and the comparatively greater information access afforded to internal auditors than external ones. However, many argue that self-regulation is insufficient and that governmental regulations are necessary in order to ensure that audits are executed by independent bodies. Several regulations exist, such as the EU's General Data Protection Regulation [57] and Germany's Network Enforcement Act [19]. There are also ongoing efforts, such as the push to review Section 230 of the U.S. Communications Decency Act [1].

Designing such regulations and auditing procedures remains a challenging problem, in part due to the number of stakeholders. For one, there are various legal and social obstacles facing regulations [45, 17, 9, 54], including concerns that regulations might damage free speech or public discourse; violate personal rights or privacy; transfer agency away from users to technology companies or governmental bodies; draw subjective lines between acceptable and unacceptable behavior; or set precedents that are difficult to reverse. In light of the thriving exchange of goods between users, platforms, advertisers and influencers that is facilitated by social media, many also fear that regulations may hurt innovation, lead to worse personalization, or block revenue sources [32].

Current efforts to audit content moderation generally focus on specific issues, such as whether content is inappropriate (e.g., posts that contain hate speech or bullying [7, 26]); discriminatory (e.g., race-based advertising [4, 62, 63, 44]); divisive (e.g., comment ranking algorithms that favor polarizing comments [61]); insulating (e.g., filter bubbles [34]), or misleading (e.g., fake news [49, 23, 25]). These works generally use one of the following strategies: increasing content diversity (e.g., adding heterogeneity to recommendations [16, 38]); drawing a line in the sand (e.g., determining whether discrimination has occurred by thresholding the difference between two proportions [24]); or finding the origin of the content (e.g., reducing fake news by whitelisting news sources [9]). In this work, we provide a general procedure such that, given a regulation in counterfactual form, an auditor can test whether the platform's filtering algorithm is compliant. Our aim is to audit with respect to the outcome of interest in order to avoid unwanted side effects. Of particular note is that the proposed procedure does not require access to users or their personal data.

We also consider how an audit affects the platform's ability to maximize an objective function as well as the content that the platform is incentivized to filter for a user. Our formulation is an instance of constrained optimization and bears resemblance to robust optimization [11, 67]. For instance, our definition of the cost of regulation mirrors the "price" of robustness studied in other works [13, 12]. Similarly, the performance-regulation trade-off that we discuss echoes the trade-offs that appears in other problems in which there are fairness [12, 33, 42] and privacy [20, 39] constraints. Our findings that there are conditions under which content diversity aligns the interests of regulators and platforms adds to the conversation on presenting different viewpoints on social media. For instance, Levy [48] finds that users respond well when presented with multiple political viewpoints—which Levy terms counter-attitudinal content—even ones from an opposing political party. Increasing content diversity is also at the heart of other methods, including those for bursting filter bubbles [16] or improving comment ranking [37]. It is worth a note that our definition of content diversity captures two notions: diversity in the topics as well as diversity in the viewpoints on each topic.

As a final remark, our analysis has parallels with differential privacy [29, 30] in that it compares outcomes under different interventions [66]. However it differs in the techniques required to study similarity. Our work also touches on aspects of but is distinct from social learning and opinion dynamics [27, 6, 3, 53] in that we study how information affects the beliefs of individuals.

## Acknowledgments and Disclosure of Funding

We would like to thank our anonymous reviewers and area chair for their time and suggestions. We would also like to thank the many people who provided feedback along the way, including but certainly not limited to Martin Copenhaver, Hussein Mozannar, Aspen Hopkins, Divya Shanmugam, and Zachary Schiffer. This work was supported in parts by the MIT-IBM project on "Representation Learning as a Tool for Causal Discovery", the NSF TRIPODS Phase II grant towards Foundations of Data Science Institute, the Chyn Duog Shiah Memorial Fellowship, and the Hugh Hampton Young Memorial Fund Fellowship.

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
