# Appendix

## A  Toy example

In this section, we provide and expand upon a toy example. Recall that the inputs $\mathbf{x}$ and $\mathbf{x}'$ need not correspond to real users but could instead represent hypothetical users.

**Example 5.** *Suppose that the regulatory guideline requires that users in the same geographical location receive similar weather forecasts. This can be written as "the weather forecasts that are selected by $\mathcal{F}$ should be similar for all users in the same geographical location", and $\mathcal{S}$ could be a randomly generated set of user pairs, where each pair corresponds to two (hypothetical) users in the same geographical location, and $\mathcal{S}$ could contain pairs across many locations.*

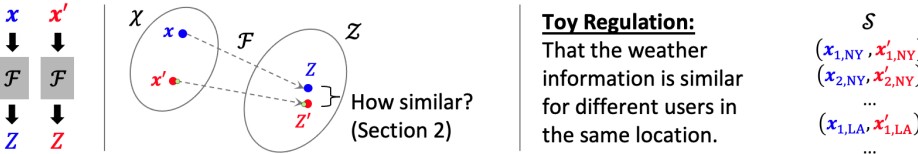

Figure 1: Visualization of toy example.

Figure 1 visualizes counterfactual regulations. In the left-most panel, a filtering algorithm $\mathcal{F}$ takes in counterfactual inputs $\mathbf{x}$ and $\mathbf{x}'$ and produces the content $Z$ and $Z'$. The middle panel visualizes this relationship graphically. Because a counterfactual regulation requires that $\mathcal{F}$ behave similarly under $\mathbf{x}$ and $\mathbf{x}'$, the regulation is effectively requiring that content $Z$ and $Z'$ are sufficiently similar (or, graphically, that they are close in $\mathcal{Z}$). The question of how to quantify "similarity" is addressed in Section 2.1. The toy example in Example 5 is illustrated in the right-most panel. Requiring that the weather information is similar for users in the same location can be tested by randomly selecting pairs of users $(\mathbf{x}, \mathbf{x}')$ in the same location, placing these pairs in $\mathcal{S}$, then running the audit over $\mathcal{S}$.

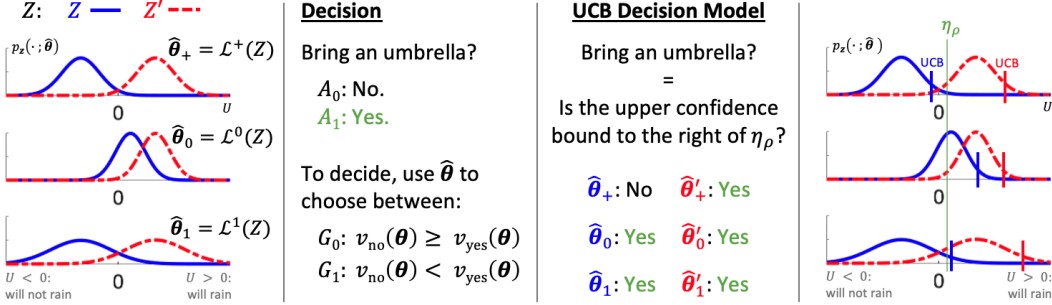

Figure 2:  Understanding the role of the MVUE (see Section 4.2).

Figure 2 visualizes the intuition behind the MVUE discussed in Section 4.2. Specifically, it illustrates why the MVUE $\mathcal{L}^+$ corresponds to the user whose decisions are most sensitive to $Z$. Suppose that a user's content $Z$ contains forecasts about the chance of rain and the user is deciding whether to bring an umbrella. Suppose the content in $Z = \mathcal{F}(\mathbf{x})$ reflects the actual chance of rain while $Z' = \mathcal{F}(\mathbf{x}')$ contains disproportionately more content suggesting that it will rain. Perhaps $Z$ is shown to children while $Z'$ is shown to adults to encourage them to buy umbrellas. Let there be three hypothetical users with estimators $\mathcal{L}^+$, $\mathcal{L}^1$, and $\mathcal{L}^2$, as indicated in the left-most panel. (As discussed in Section 4.2, every user ingests their content differently. An estimator $\mathcal{L}$ is simply a mapping from content $Z$ to the user's belief. In this toy example, we study three hypothetical users.)

In the left-most panel, each plot visualizes one of the user's belief $p_{\mathbf{z}}(\cdot; \hat{\boldsymbol{\theta}})$ about whether it will rain $U$ given $Z$ (in solid blue) or given $Z'$ (in dashed red), where $U < 0$ suggests that it will not rain and $U > 0$ suggests that it will rain. $\mathcal{L}^+$ is the MVUE, $\mathcal{L}^1$ is a biased estimator (it is biased to the right such that the user tends to believe it will rain today no matter what the forecasts say), and $\mathcal{L}^2$ is

an unbiased estimator with higher variance than $\mathcal{L}^+$ (the user does not put much confidence in the forecasts, so its belief is less "peaky" than the MVUE's).

In the second panel, we write the decision of whether to bring an umbrella in terms of the setup in Section 4.2. Specifically, if the user knew that the true chance of rain as given by $\boldsymbol{\theta}$, they would bring an umbrella if $v_{\text{yes}}(\boldsymbol{\theta}) > v_{\text{no}}(\boldsymbol{\theta})$ and would not bring an umbrella, otherwise, where $v_i$ denotes the value that the user places on each option. For example, $v_i$ may balance the user's dislike of carrying an umbrella with the user's dislike of walking in the rain, and $v_i$ may differ across individuals.

The third panel explains how the user would make a decision under the upper confidence bound (UCB) decision model, a popular model in the bandit literature [18]. Here $\boldsymbol{\theta}$ captures the reward and sampling history of the bandit (i.e., the past experiences of a user with respect to rain and weather forecasts), and $v_i(\boldsymbol{\theta})$ would give the UCB of arm $i$ (i.e., of the choices to and not to bring an umbrella). As written in the third panel, under the UCB decision model, the user would choose to bring an umbrella if the UCB of their belief is to the right of some threshold $\eta_\rho$ (for details on $\eta_\rho$, see Section 4.2) and would not bring an umbrella, otherwise.

In order to understand what decision each of the three users corresponding to $\mathcal{L}^+$, $\mathcal{L}^1$, and $\mathcal{L}^2$ would do, examine the fourth (right-most) panel. Let the threshold $\eta_\rho$ be given by the thin vertical line, as marked. Let the UCB for $Z$ and $Z'$ be given by the blue and red thick lines, as indicated in the top-most plot (the blue line is always to the left of the red line). We see that, for this choice of $\eta_\rho$, the MVUE would not choose to bring an umbrella under $Z$ but would choose to do so under $Z'$. We also see that the users corresponding to $\mathcal{L}^1$ and $\mathcal{L}^2$ would choose to bring umbrellas under both $Z$ and $Z'$. These choices are also written in the third-panel from the left.

The goal of Figure 2 is to provide intuition for why the MVUE corresponds to the "most gullible user": the hypothetical user whose decisions are most affected by their content. Recall that $Z'$ indicates that it is more likely to rain than $Z$. As illustrated in the example, the MVUE is the only estimator among the three for which the user's decision is different when shown $Z$ versus $Z'$, whereas the users corresponding to the other estimators are less affected by the content that they see: their decisions remain the same under $Z$ and $Z'$. This example confirms the discussion in Section 4.2 that the decisions of the MVUE are more sensitive to whether the content is $Z$ or $Z'$ than the decisions of other users (i.e., other estimators). Therefore, if we wish to enforce similarity between users' decision-making behavior under $Z$ and $Z'$—or, equivalently, under the inputs $\mathbf{x}$ and $\mathbf{x}'$—then the MVUE provides an "upper bound" on the sensitivity of users' decisions to their content.

For intuition on why the MVUE is the "most gullible user", recall that the MVUE is the unbiased estimator with the lowest variance. Suppose that a user's estimate $\mathcal{L}(Z)$ differs from $\mathcal{L}^+(Z)$. By definition, this estimate is biased or has higher variance. When biased, the user's estimate is consistently pulled by some factor other than $Z$. For example, a user who remains pro-vaccine no matter what content they see has a biased estimator. When the user's estimate has higher variance than the MVUE's, it is an indication that the user places less confidence than the MVUE in what they glean from $Z$. For example, the user could be skeptical of what they see on social media or scrolling very quickly and only reading headlines. In this way, the MVUE corresponds to the user the user who "hangs on every word"—whose decisions are most affected by their content $Z$.

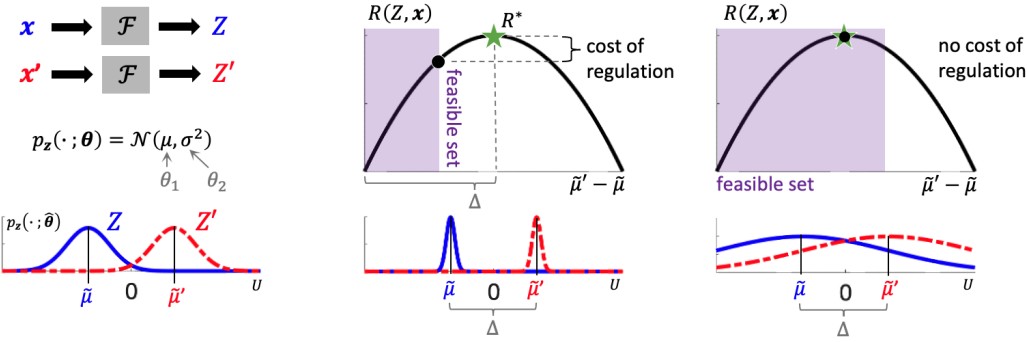

Figure 3: Visualizing the cost of regulation and the connection to content diversity.

Figure 3 visualizes the cost of regulation and illustrates why increasing content diversity can reduce the cost of regulation. In this example, suppose that $p_{\mathbf{z}}(\cdot; \boldsymbol{\theta})$ are 1-D Gaussian, as in Example 4. We examine how different choices of $(Z, Z')$ affect (a) the feasible set and (b) the platform's cost of regulation. In the left-most panel, we re-iterate that $Z = \mathcal{F}(\mathbf{x})$ and $Z' = \mathcal{F}(\mathbf{x}')$. As stated above, we assume that $\Theta$ parameterizes the family of 1-D Gaussian. The bottom of the left-most panel provides an example of how an estimator $\mathcal{L}$ would behave if given $Z$ and $Z'$ where $Z$ is a 1-D Gaussian centered to the left of 0, $Z'$ is a 1-D Gaussian centered to the right of 0, and both have the same variance $\sigma^2$. For this example, we assume that the variance $\sigma^2$ is the same for $Z$ and $Z'$. Specifically, the distribution under $\hat{\boldsymbol{\theta}} = \mathcal{L}(Z)$ is plotted in solid blue while the distribution under $\hat{\boldsymbol{\theta}} = \mathcal{L}(Z')$ is plotted in dashed red. For the purposes of this example, one can assume that $\mathcal{L} = \mathcal{L}^+$ is the MVUE, and we denote the estimate of $\boldsymbol{\theta}$ by $\tilde{\boldsymbol{\theta}}$ to be consistent with the notation in the main text.

The middle and right-most panels visualize the cost of regulation under two different choices of $(Z, Z')$. In the middle panel, $Z$ and $Z'$ are chosen such that the distributions $p(\cdot; \tilde{\boldsymbol{\theta}})$ under $Z$ and $Z'$ are given in the middle-bottom. Specifically, $\tilde{\mu}$ and $\tilde{\mu}'$ are $\Delta$ apart, and the variance $\sigma^2$ is fairly small.

Suppose that, as in Example 4, $R$ is a function of the means (i.e., of $\tilde{\mu}$ and $\tilde{\mu}'$) but not of the variance $\sigma^2$. Furthermore, suppose that the platform maximizes its reward $R$ when $\tilde{\mu}' - \tilde{\mu} = \Delta$, as visualized by the green star in the top middle plot. As explained in Section 5, a regulation restricts the platform's choice of feeds (or collections of type-$T$ content) from $\mathcal{Z}$ to the feasible set $\mathcal{Z}(\mathcal{S})$, which is a subset of $\mathcal{Z}$. In the top middle panel, we visualize the feasible set in purple. Specifically, for the given choice of $\sigma^2$ (as plotted in the bottom middle), the feasible set does not include $\tilde{\mu}' - \tilde{\mu} = \Delta$. The maximum reward that the platform can achieve under the regulation for this choice of $\sigma^2$ is indicated by the black dot, and the vertical distance between the green star and black dot is the cost of regulation.

However, if the platform still chooses $Z$ and $Z'$ such that $\tilde{\mu}' - \tilde{\mu} = \Delta$ but increase the variance $\sigma^2$, the story is different. In the right-most panel, we show that, for a larger $\sigma^2$, the feasible set expands to include $\tilde{\mu}' - \tilde{\mu} = \Delta$. As such, the reward-maximizing solution is contained within the feasible set, and there is no cost of regulation. In this way, adding a sufficient amount of content diversity can reduce the cost of regulation, thereby allowing the platform to achieve high reward while also complying with the regulation.

Mathematically, this example is explained by Theorem 3 and discussed in Example 4. Expanding on the discussion in Example 4, recall that, when $\boldsymbol{\theta} = (\mu, \sigma^2)$, $\Theta = \mathbb{R} \times \mathbb{R}_{\geq 0}$, $\mathcal{P} = \{\mathcal{N}(\mu, \sigma^2) : (\mu, \sigma^2) \in \Theta\}$ is the family of 1-D Gaussian distributions, and $R$ is a function of $\mu$ but not $\sigma^2$ (as in Figure 3), then $I((\mu, \sigma^2)) = \mathrm{diag}(\sigma^{-2}, \sigma^{-4}/2)$, $\bar{\boldsymbol{\theta}}_{\Omega} = (0, 1)$, and

$$\mathbf{v}^{\top} I(\boldsymbol{\theta} + \kappa \bar{\boldsymbol{\theta}}_{\Omega})\mathbf{v} = v_1^2/(\sigma^2 + \kappa) + v_2^2/(2(\sigma^2 + \kappa)^2). \tag{5}$$

This quantity becomes very small if $\kappa$ is very large. It turns out that making this quantity small is precisely what we want and that making $\kappa$ large is the same as increasing the content diversity because $\boldsymbol{\theta} + \kappa \bar{\boldsymbol{\theta}}_{\Omega} = (\mu, \sigma^2 + \kappa)$. To see this connection, recall that the pair $Z$ and $Z'$ passes regulation if $(\mathcal{L}^+(Z) - \mathcal{L}^+(Z'))^{\top} I(\mathcal{L}^+(Z))(\mathcal{L}^+(Z) - \mathcal{L}^+(Z')) < \frac{2}{m}\chi_r^2(1 - \epsilon)$, Therefore, given content $Z^*$ for which $\mathcal{L}^+(Z^*) = \tilde{\boldsymbol{\theta}}^*$ and $R(Z^*, \mathbf{x})$ is the maximum achievable reward, one can create a new feed (or collection of type-$T$ content) $Z$ that passes regulation by taking $Z^*$ and increasing its content diversity such that $\mathcal{L}^+(Z) = (\mu^*, (\sigma^*)^2 + \kappa)$. By (5), the quantity $(\mathcal{L}^+(Z) - \mathcal{L}^+(Z'))^{\top} I(\mathcal{L}^+(Z))(\mathcal{L}^+(Z) - \mathcal{L}^+(Z'))$ can be made arbitrarily small by taking $\kappa$ to be large, which means that $Z$ is in the feasible set. Moreover, since $R$ does not depend on the variance, $R(Z, \mathbf{x}) = R(Z^*, \mathbf{x})$. Note that this example is highly simplified as an illustration, but the intuition that it provides holds more generally.

## B   Technical details

Recall that the Fisher information matrix $I(\boldsymbol{\theta}) \in \mathbb{R}^{r \times r}$ is a positive semi-definite matrix, where the $(i, j)$-th entry is given by:

$$[I(\boldsymbol{\theta})]_{ij} = \mathbb{E}_{\mathbf{z} \sim p_{\mathbf{z}}(\cdot; \boldsymbol{\theta})} \left[ \frac{\partial}{\partial \theta_i} \log p_{\mathbf{z}}(\mathbf{z}; \boldsymbol{\theta}) \frac{\partial}{\partial \theta_j} \log p_{\mathbf{z}}(\mathbf{z}; \boldsymbol{\theta}) \right]$$

Recall that $Z$ is generated by drawing $m$ samples from $p_{\mathbf{z}}(\cdot; \boldsymbol{\theta})$, where $\boldsymbol{\theta} \in \Theta$. Recall further that $\mathcal{L} : \mathcal{Z} \to \Theta$ denotes an estimator. An estimator $\mathcal{L}$ is asymptotically normal and efficient if:

$$\sqrt{m}\left(\mathcal{L}(Z) - \boldsymbol{\theta}\right) \xrightarrow{d} \mathcal{N}(\mathbf{0}_r, I^{-1}(\boldsymbol{\theta})), \tag{6}$$

as $m \to \infty$ for all $\boldsymbol{\theta} \in \Theta$ where $I^{-1}(\boldsymbol{\theta})$ denotes the inverse of the Fisher information matrix at $\boldsymbol{\theta}$.

Lastly, let $\mathcal{P} = \{p_{\mathbf{z}}(\cdot; \boldsymbol{\theta}) : \boldsymbol{\theta} \in \Theta\}$. The regularity conditions on $\mathcal{P}$ that are discussed in Theorem 1 are stated as follows.

1. $\Theta$ is a compact and open set of $\mathbb{R}^r$.

2. Identifiability: $\mathbf{z} \overset{\text{i.i.d.}}{\sim} p_{\mathbf{z}}(\cdot; \boldsymbol{\theta})$ for $\boldsymbol{\theta} \in \Theta$ and $\boldsymbol{\theta}_1 \neq \boldsymbol{\theta}_2$ implies $p_{\mathbf{z}}(\cdot; \boldsymbol{\theta}_1)$ and $p_{\mathbf{z}}(\cdot; \boldsymbol{\theta}_2)$ are distinct.

3. Common support: The support of $p_{\mathbf{z}}(\cdot; \boldsymbol{\theta})$ is independent of $\boldsymbol{\theta} \in \Theta$.

4. Differentiability: All the second-order partial derivates of $\log p_{\mathbf{z}}(\mathbf{z}; \boldsymbol{\theta})$ with respect to $\boldsymbol{\theta}$ exist and are continuous in $\boldsymbol{\theta}$.

5. For any $\boldsymbol{\theta}_0 \in \Theta$, there exists a neighborhood of $\boldsymbol{\theta}_0$ and a function $\Pi(\mathbf{z})$, where $\mathbb{E}_{\mathbf{z} \sim p_{\mathbf{z}}(\cdot; \boldsymbol{\theta}_0)}[\Pi(\mathbf{z})] < \infty$ and

$$\left| \frac{\partial^2}{\partial \theta_i \partial \theta_j} \log p_{\mathbf{z}}(\mathbf{z}; \boldsymbol{\theta}) \right| \leq \Pi(\mathbf{z}),$$

for all $\mathbf{z} \in \mathcal{Z}$, all $\boldsymbol{\theta}$ in the neighborhood of $\boldsymbol{\theta}_0$, and $i, j \in [r]$.

6. If $\boldsymbol{\theta}^*$ is the data generating parameter:

   (a) $\frac{\partial}{\partial \theta_i} \log p_{\mathbf{z}}(\mathbf{z}; \boldsymbol{\theta}^*)$ is square integrable for all $i \in [r]$.

   (b) $\mathbb{E}_{\mathbf{z} \sim p_{\mathbf{z}}(\cdot; \boldsymbol{\theta}^*)} \left[ \frac{\partial}{\partial \theta_i} \log p_{\mathbf{z}}(\mathbf{z}; \boldsymbol{\theta}^*) \right] = 0$

   (c) The Fisher information at $\boldsymbol{\theta}^*$ satisfies:

   $$[I(\boldsymbol{\theta}^*)]_{ij} = \mathbb{E}_{\mathbf{z} \sim p_{\mathbf{z}}(\cdot; \boldsymbol{\theta}^*)} \left[ \frac{\partial}{\partial \theta_i} \log p_{\mathbf{z}}(\mathbf{z}; \boldsymbol{\theta}^*) \frac{\partial}{\partial \theta_j} \log p_{\mathbf{z}}(\mathbf{z}; \boldsymbol{\theta}^*) \right]$$

   $$= -\mathbb{E}_{\mathbf{z} \sim p_{\mathbf{z}}(\cdot; \boldsymbol{\theta}^*)} \left[ \frac{\partial^2}{\partial \theta_i \theta_j} \log p_{\mathbf{z}}(\mathbf{z}; \boldsymbol{\theta}^*) \right]$$

   (d) Invertibility: Fisher information $I(\boldsymbol{\theta}^*)$ at $\boldsymbol{\theta}^*$ is positive-definite and invertible.

7. Either all distributions in $\mathcal{P}$ are lattice distributions on the same lattice or each $p_{\mathbf{z}}(\cdot; \boldsymbol{\theta}) \in \mathcal{P}$ has a component such that, for a constant $k$ that is independent of $\boldsymbol{\theta}$, the $k$-fold convolution has a bounded density with respect to the Lebesgue measure.

8. For all $\boldsymbol{\theta} \in \Theta$, there exists an unbiased estimator $\mathcal{L}$ such that $\mathbb{E}[|\mathcal{L}(Z)|] < \infty$.

9. $\Theta$ is a convex set.

There are variations on these regularity conditions, and we refer the reader to other works for further details [5, 46, 50]. The compactness requirement in Condition 1 and the continuity requirement in Condition 4 ensure the existence of the MLE. The remaining statements in Conditions 1 through 6 ensure the asymptotic normality of the MLE. Conditions 7–8 ensure the asymptotic normality of the MVUE (cf. [59] for details). Condition 9 ensures the existence of $I((\boldsymbol{\theta} + \boldsymbol{\theta}')/2)$ for $\boldsymbol{\theta}, \boldsymbol{\theta}' \in \Theta$, and this condition can be relaxed by providing a slightly different statement of Theorem 1 (e.g., letting $\boldsymbol{\theta}^*$ be $\boldsymbol{\theta}(\mathbf{x})$ or $\boldsymbol{\theta}(\mathbf{x}')$).

## C   Proofs

### C.1   Theorem 1

**Theorem 1.** *Consider* (1). *Let* $\boldsymbol{\theta}^* = (\boldsymbol{\theta}(\mathbf{x}) + \boldsymbol{\theta}(\mathbf{x}'))/2$. *Suppose that* $\mathbf{z}_i$ *and* $\mathbf{z}_i'$ *are drawn i.i.d. from* $p(\cdot; \boldsymbol{\theta}(\mathbf{x}))$ *and* $p(\cdot; \boldsymbol{\theta}(\mathbf{x}'))$, *respectively, for all* $i \in [m]$ *and* $\mathcal{P} = \{p_{\mathbf{z}}(\cdot; \boldsymbol{\theta}) : \boldsymbol{\theta} \in \Theta\}$ *is a regular exponential family that meets the regularity conditions stated in Appendix B. If* $\hat{H}$ *is defined as:*

$$\hat{H} = H_1 \iff (\mathcal{L}^+(Z) - \mathcal{L}^+(Z'))^\top I(\boldsymbol{\theta}^*)(\mathcal{L}^+(Z) - \mathcal{L}^+(Z')) \geq \frac{2}{m} \chi_r^2(1 - \epsilon),$$

*then* $\mathbb{P}(\hat{H} = H_1 | H = H_0) \leq \epsilon$ *as* $m \to \infty$. *If* $r = 1$, *then* $\hat{H}$ *is the UMPU test as* $m \to \infty$, *i.e.,* $\lim_{m \to \infty} \mathbb{P}(\hat{H} = H_0 | H = H_1) = \alpha^*(\epsilon)$.

*Proof.* The regularity conditions required for Theorem 1 are stated in Appendix B. The definition of asymptotic normality and efficiency is also given in Appendix B.

Under the regularity conditions, we have three results. First, under Conditions 1 and 4, the MLE exists and, from Conditions 1-6, it is asymptotically normal and efficient [5, 46, 50]. Second, under Conditions 1-8, the MVUE exists and is also asymptotically normal and efficient [59]. Third, Condition 9 ensures the existence of $I((\boldsymbol{\theta} + \boldsymbol{\theta}')/2)$ for $\boldsymbol{\theta}, \boldsymbol{\theta}' \in \Theta$, and this condition can be relaxed by providing a slightly different statement of Theorem 1 (e.g., letting $\boldsymbol{\theta}^*$ be $\boldsymbol{\theta}(\mathbf{x})$ or $\boldsymbol{\theta}(\mathbf{x}')$).

By the second result,

$$\sqrt{m}(\mathcal{L}^+(Z) - \boldsymbol{\theta}(\mathbf{x})) \xrightarrow{d} \mathcal{N}(\mathbf{0}_r, I^{-1}(\boldsymbol{\theta}(\mathbf{x})))$$

as $m \to \infty$, where $\mathbf{z}_i \stackrel{\text{i.i.d.}}{\sim} p(\cdot; \boldsymbol{\theta})$ and $Z = (\mathbf{z}_1, \ldots, \mathbf{z}_m)$. Therefore, as $m \to \infty$,

$$\sqrt{m}(\mathcal{L}^+(Z) - \boldsymbol{\theta}(\mathbf{x}) - \mathcal{L}^+(Z') + \boldsymbol{\theta}(\mathbf{x}')) \xrightarrow{d} \mathcal{N}(\mathbf{0}_r, I^{-1}(\boldsymbol{\theta}(\mathbf{x})) + I^{-1}(\boldsymbol{\theta}(\mathbf{x}'))) \tag{7}$$

Recall the hypothesis test (1) from Section 2.2. When $H = H_0$, $\boldsymbol{\theta}(\mathbf{x}) = \boldsymbol{\theta}(\mathbf{x}') = \boldsymbol{\theta}^*$. Therefore, by (7),

$$\sqrt{m}(\mathcal{L}^+(Z) - \mathcal{L}^+(Z')) \xrightarrow{d} \mathcal{N}(\mathbf{0}_r, 2I^{-1}(\boldsymbol{\theta}^*)) \tag{8}$$

as $m \to \infty$ when $H = H_0$, which implies that, as $m \to \infty$, the two-sample, two-sided hypothesis test in (1) becomes a two-sample, one-sided test of on the mean of a multivariate Gaussian random variable. Under (8),

$$(\mathcal{L}^+(Z) - \mathcal{L}^+(Z'))^\top I(\boldsymbol{\theta}^*)(\mathcal{L}^+(Z) - \mathcal{L}^+(Z')) \sim \frac{2}{m}\chi_r^2$$

Therefore, if $\hat{H}$ satisfies:

$$\hat{H} = H_1 \iff (\mathcal{L}^+(Z) - \mathcal{L}^+(Z'))^\top I(\boldsymbol{\theta}^*)(\mathcal{L}^+(Z) - \mathcal{L}^+(Z')) \geq \frac{2}{m}\chi_r^2(1 - \epsilon) \tag{9}$$

then $\hat{H}$ has a FPR $\leq \epsilon$, as desired.

Although $\hat{H}$ is not necessarily the UMPU for $r > 1$, it is well known that it is the UMPU test of size $\epsilon$ for the univariate Gaussian case, i.e., when $r = 1$ (cf. Section 8.3 of [21]).

One may have noticed that the hypothesis test in (9) (and (3)) uses $Z$ and $Z'$ to choose between $H_0$ and $H_1$, whereas in the original problem statement in Section 2, the hypothesis test uses $\mathcal{D}$ and $\mathcal{D}'$. In other words, decision robustness requires that one cannot determine whether $\mathbf{x} \neq \mathbf{x}'$—or, equivalently, $\boldsymbol{\theta}(\mathbf{x}) \neq \boldsymbol{\theta}(\mathbf{x}')$—from $\mathcal{D}$ and $\mathcal{D}'$ for any $\mathcal{Q}$. Although $\mathcal{D}, \mathcal{D}'$, and $\mathcal{Q}$ do not appear in the analysis above, the test in (9) ensures (approximate asymptotic) decision robustness by designing a test that works for all $\mathcal{Q}$ and consequent decisions $\mathcal{D}$ and $\mathcal{D}'$.

To see this, we first note that, if expressed as a Markov chain, the random variables of interest would be written as $\mathbf{x} \to \boldsymbol{\theta} \to Z \to \mathcal{D}$. By the data processing inequality, any test that uses $Z$ is stronger (i.e., has a higher TPR and lower FPR) than the corresponding test using $\mathcal{D}$. Intuitively, since $\mathcal{D}$ is determine by $Z$ and $\mathcal{D}'$ from $Z'$, if one cannot determine whether $\mathbf{x} \neq \mathbf{x}$ (or $\boldsymbol{\theta}(\mathbf{x}) \neq \boldsymbol{\theta}(\mathbf{x}')$) from $Z$ and $Z'$, then one cannot do any better given $\mathcal{D}$ and $\mathcal{D}'$ for any $\mathcal{Q}$. We can therefore conclude that the guarantees of (9) hold for all $\mathcal{Q}$ and the original hypothesis test in (1). $\square$

Note that the regularity conditions required in Theorem 1 are fairly mild. Recall that exponential families capture a broad class of distributions of interest. In particular, they are the only families of distributions that have finite-dimensional sufficient statistics, and a distribution almost always belongs to an exponential family if it has a conjugate prior. Regular exponential families are canonical exponential families if the natural parameter space is an open set in $\Theta$. The remaining regularity conditions are common and often implicitly assumed in discussions of the MVUE or MLE.

**Alternate result**. We would like to remark that the audit could be modified to use the MLE instead of the MVUE. Using the MLE would provide the same guarantee as Theorem 1. In fact, it would require fewer conditions, as follows.

**Theorem C.1.** *Consider (1). Let $\boldsymbol{\theta}^* = (\boldsymbol{\theta}(\mathbf{x}) + \boldsymbol{\theta}(\mathbf{x}'))/2$. Suppose that $\mathbf{z}_i$ and $\mathbf{z}_i'$ are drawn i.i.d. from $p(\cdot; \boldsymbol{\theta}(\mathbf{x}))$ and $p(\cdot; \boldsymbol{\theta}(\mathbf{x}'))$, respectively, for all $i \in [m]$ and $\mathcal{P} = \{p_{\mathbf{z}}(\cdot; \boldsymbol{\theta}) : \boldsymbol{\theta} \in \Theta\}$ meets Conditions 1 through 6 stated in Appendix B. If $\hat{H}$ is defined as:*

$$\hat{H} = H_1 \iff (\mathcal{L}^+(Z) - \mathcal{L}^+(Z'))^\top I(\boldsymbol{\theta}^*)(\mathcal{L}^+(Z) - \mathcal{L}^+(Z')) \geq \frac{2}{m}\chi_r^2(1-\epsilon), \qquad (10)$$

*then $\mathbb{P}(\hat{H} = H_1 | H = H_0) \leq \epsilon$ as $m \to \infty$. If $r = 1$, then $\hat{H}$ is the UMPU test as $m \to \infty$.*

Therefore, one may wish to use the MLE instead of the MVUE in Algorithm 1. The reason that one may wish to use the MVUE is because it provides another guarantee (cf. Proposition 2).

## C.2 Proposition 2

**Proposition 2.** *Consider (4). Let $G \in \{G_0, G_1\}$ denote the true (unknown) hypothesis. Suppose that $v_0, v_1 : \Theta \to \mathbb{R}$ are affine mappings and there exists $\mathbf{u} : \mathbb{R}^n \to \mathcal{U}$ such that, for $\mathcal{F}(\mathbf{x}) = \{\mathbf{z}_1, \ldots, \mathbf{z}_m\}$, one can write $\mathbf{u}(\mathbf{z}_i) \sim p_{\mathbf{u}}(\cdot; v_1(\boldsymbol{\theta}(\mathbf{x})) - v_0(\boldsymbol{\theta}(\mathbf{x})))$ for all $i \in [m]$. Then, if the UMP test with a maximum FPR of $\rho$ exists, it is given by the following decision rule: reject $G_0$ (choose $A_1$) when the minimum-variance unbiased estimate $\tilde{\boldsymbol{\theta}} = \mathcal{L}^+(Z)$ satisfies $v_1(\tilde{\boldsymbol{\theta}}) - v_0(\tilde{\boldsymbol{\theta}}) > \eta_\rho$ where $\mathbb{P}(v_1(\tilde{\boldsymbol{\theta}}) - v_0(\tilde{\boldsymbol{\theta}}) > \eta_\rho | G = G_0) = \rho$; otherwise, accept $G_0$ (choose $A_0$).*

*Proof.* We begin with a result from Ghobadzadeh et al. [36].

**Lemma C.2** ([36], Theorem 1). *Consider a one-sided binary composite hypothesis test of $\bar{G}_0 : \mathbf{w} \sim p_{\mathbf{w}}(\cdot; \gamma), \gamma \leq \gamma_b$ against $\bar{G}_1 : \mathbf{w} \sim p_{\mathbf{w}}(\cdot; \gamma), \gamma > \gamma_b$, where $\gamma_b$ is known. Let $\Gamma_0 = \{\gamma : \gamma \leq \gamma_b\}$, $\Gamma_1 = \{\gamma : \gamma > \gamma_b\}$, and $\Gamma = \Gamma_0 \cup \Gamma_1$. Let $\rho$ be the maximum allowable false positive rate. If the uniformly most powerful (UMP) test exists, then it is defined by the following decision rule: reject $\bar{G}_0$ when the minimum variance unbiased estimator (MVUE) of $\gamma \in \Gamma$, denoted by $\tilde{\gamma}^+$, satisfies $\tilde{\gamma}^+ > \gamma_\rho$, where $P(\tilde{\gamma}^+ > \gamma_\rho | H = \bar{G}_0) = \rho$.*

Our result follows directly from two observations. First, since $v_0, v_1$ are affine, if $\tilde{\boldsymbol{\theta}}$ is the MVUE of $\boldsymbol{\theta}$, then $v_1(\tilde{\boldsymbol{\theta}}) - v_0(\tilde{\boldsymbol{\theta}})$ is also the MVUE of $v_1(\boldsymbol{\theta}) - v_0(\boldsymbol{\theta})$. Second, our setting is equivalent to that in Lemma C.2 with the substitutions $\gamma = v_1(\boldsymbol{\theta}) - v_0(\boldsymbol{\theta})$, $\gamma_b = 0$, and $\mathbf{w} = \mathbf{u}(\mathbf{z})$. $\square$

## C.3 Theorem 3

Recall from footnote 10 that $R$ can be time-varying. For example,, the platform's revenue sources may change with time. As long as the time-varying objective function $R^{(t)}$ satisfies the conditions in Theorem 3 at every time step, then the result holds unchanged at every time step.

**Theorem 3.** *Suppose there exists $\Omega \subset [r]$ where $1 < |\Omega| < r$ such that $R(\boldsymbol{\theta}_1, \mathbf{x}) = R(\boldsymbol{\theta}_2, \mathbf{x})$ if $\theta_{1,i} = \theta_{2,i}$ for all $i \notin \Omega$. Suppose that, for any $\boldsymbol{\theta} \in \Theta$, $\beta > 0$ and $\mathbf{v} \in \mathbb{R}^r$, there exist a vector $\bar{\boldsymbol{\theta}}_\Omega$ where $\bar{\theta}_{\Omega,i} = 0$ for all $i \notin \Omega$ and a constant $\kappa > 0$ such that $\mathbf{v}^\top I(\boldsymbol{\theta} + \kappa \bar{\boldsymbol{\theta}}_\Omega)\mathbf{v} < \beta$ and $\boldsymbol{\theta} + \kappa \bar{\boldsymbol{\theta}}_\Omega \in \Theta$. Then, if $m < \infty$, there exists a $\mathcal{Z}$ such that the cost of regulation for $\mathbf{x}$ under Algorithm 1 is 0.*

*Proof.* Let $Z_* \in \arg\max_{W \in \mathcal{Z}} R(W, \mathbf{x})$ be a reward-maximizing solution and $\tilde{\boldsymbol{\theta}}_* = \mathcal{L}^+(Z_*)$. Similarly, let $Z_*' \in \arg\max_{W \in \mathcal{Z}} R(W, \mathbf{x}')$ and $\tilde{\boldsymbol{\theta}}_*' = \mathcal{L}^+(Z_*')$. Recall that, under the statement conditions, there exists a vector $\bar{\boldsymbol{\theta}}_\Omega$ where $\bar{\theta}_{\Omega,i} = 0$ for all $i \notin \Omega$ and a constant $\kappa > 0$ such that $\mathbf{v}^\top I(\boldsymbol{\theta} + \kappa \bar{\boldsymbol{\theta}}_\Omega)\mathbf{v} < \beta$ and $\boldsymbol{\theta} + \kappa \bar{\boldsymbol{\theta}}_\Omega \in \Theta$ for any $\boldsymbol{\theta} \in \Theta$, $\beta > 0$, and $\mathbf{v} \in \mathbb{R}^r$. Let us take $\boldsymbol{\theta} = \tilde{\boldsymbol{\theta}}_* = \mathcal{L}^+(Z_*)$, $\beta = \frac{2}{m}\chi_r^2(1-\epsilon)$, and $\mathbf{v} = \tilde{\boldsymbol{\theta}}_*' - \tilde{\boldsymbol{\theta}}_*$. Finally, let $\tilde{\boldsymbol{\theta}} = \tilde{\boldsymbol{\theta}}_* + \bar{\boldsymbol{\theta}}_\Omega$, where $\bar{\boldsymbol{\theta}}_\Omega$ is defined as given in the statement. Then,

$$(\tilde{\boldsymbol{\theta}}_*' - \tilde{\boldsymbol{\theta}}_*)^\top I(\tilde{\boldsymbol{\theta}}_* + \bar{\boldsymbol{\theta}}_\Omega)(\tilde{\boldsymbol{\theta}}_*' - \tilde{\boldsymbol{\theta}}_*) < \frac{2}{m}\chi_r^2(1-\epsilon).$$

Letting $\tilde{\boldsymbol{\theta}}' = \tilde{\boldsymbol{\theta}}_*' + \bar{\boldsymbol{\theta}}_\Omega$ and recalling $\tilde{\boldsymbol{\theta}} = \tilde{\boldsymbol{\theta}}_* + \bar{\boldsymbol{\theta}}_\Omega$ gives

$$(\tilde{\boldsymbol{\theta}}' - \tilde{\boldsymbol{\theta}})^\top I(\tilde{\boldsymbol{\theta}})(\tilde{\boldsymbol{\theta}}' - \tilde{\boldsymbol{\theta}}) < \frac{2}{m}\chi_r^2(1-\epsilon),$$

which implies that, as long as $\mathcal{Z}$ is large enough such that there exist $Z$ and $Z'$ such that $\mathcal{L}^+(Z) = \tilde{\boldsymbol{\theta}}$ and $\mathcal{L}^+(Z') = \tilde{\boldsymbol{\theta}}'$, then both $Z$ and $Z'$ comply with the regulation. In other words, as long as $\mathcal{Z}$ contains content that is expressive enough, we know that $Z, Z' \in \mathcal{Z}(\mathcal{S})$.

It remains to show that the cost of regulation is $0$. To do so, we show that $Z$, which is in the feasible set, achieves the maximum reward $\max_{W \in \mathcal{Z}} R(W, \mathbf{x}) = R(\tilde{\boldsymbol{\theta}}_*, \mathbf{x})$. That the cost of regulation is $0$ follows from the fact that $R(\tilde{\boldsymbol{\theta}}_*, \mathbf{x}) = R(\tilde{\boldsymbol{\theta}}_* + \bar{\boldsymbol{\theta}}_\Omega, \mathbf{x})$ because $\bar{\theta}_{\Omega,i} = 0$ for $i \notin \Omega$. $\qquad\square$

## D    Additional discussion

### D.1    Two remarks

Recall that we made two simplifications in the main text for readability. As noted in the main text, these simplifications do not change our main findings, which implies that our results hold under conditions more general than those given in the main text.

First, recall from footnote 10 that $R$ can be time-varying, i.e., let the objective function that the platform wishes to maximize at time step be given by $R^{(t)}$. This allows our analysis to accommodate settings in which the platform's objectives (e.g., revenue sources) change with time. Allowing the objective function to vary in time does not change our conclusions. Our findings with respect to $R$ appear in Theorem 3 and the discussion that follows. As apparent in the proof of Theorem 3, adopting a time-varying $R^{(t)}$ leads to the same result as long as $R^{(t)}$ satisfies the conditions in the theorem statement at the time step $t$ of interest.

Second, recall from footnote 9 that a user's beliefs are often influenced by information other than the content $Z$ that they view on the social media platform. For instance, a user's beliefs may depend on conversations that they have offline or on their previous beliefs. One could incorporate this into our setup by letting $J \in \mathcal{J}$ denote any information other than $Z$ that the user uses to form or update their beliefs and the estimator $\mathcal{L} : \mathcal{Z} \times \mathcal{J} \to \Theta$ denote the user's learning behavior (that incorporates information both on and off the platform) such that the user's belief after viewing $Z$ and observing information $J$ is given by $\mathcal{L}(Z, J)$.

The outside information $J$ does not affect our results because it can be absorbed into $\mathcal{L}$. That is, because we are only interested in how the user's beliefs are affected when the user is shown $Z'$ instead of $Z$, and vice versa, $J$ can effectively be ignored. Another way to see this is by recalling the definition of decision robustness. Decision robustness supposes that there are two identical (hypothetical) users, one of whom is shown $Z$ and the other $Z'$. For the purpose of comparing the outcomes under $Z$ and $Z'$, $J$ could be treated as part of the original identical users.

### D.2    Impact and additional considerations

Our hope is that this work can contribute to the ongoing conversations about social media and its governance. In light of the difficulties in designing and enforcing a regulation, we focus on the latter half of the process by proposing an auditing procedure. Auditing social media remains a challenging topic because changes to the ecosystem can have far-reaching consequences. As such, we sought to consider the various stakeholders in the system.

In particular, we studied our framework from the perspectives of the auditor, platform, and user. The proposed test focuses on a given pair of inputs. We made this choice intentionally to prevent issues that often arise when a regulatory test focuses only on average behavior, which can sometimes result in good outcomes for most individuals but unsatisfactory outcomes for a small subset of the population (i.e., a minority group). We considered the perspective of the auditor by acknowledging the difficulties in designing regulations that are enduring and adaptable while also being precise and implementable. To this end, our main contribution is a test that translates counterfactual regulations into a principled regulatory procedure. We also consider the platform's perspective by studying how the audit affects the platform's ability to maximize some objective function (e.g., revenue, user engagement, a combination of these factors, and more). This discussion returns to the user's perspective by examining how the audit changes the content or feed that the platform is incentivized to show users with particular attention to the diversity of the user's content.

To the best of our abilities, we attempt to acknowledge and address the impact of our work by considering various perspectives of our proposal, explicitly mentioning what problems are within the scope of this work and pointing to appropriate references. However, there may be angles that we have missed. There is also the potential to misuse the proposed framework. For instance, if a platform decided to adopt our procedure as a self-regulatory measure, the outcome would depend on how seriously the platform engages in conversations on designing the counterfactual inputs. Another potential misuse would be adversarially designing the features that represent content such that the regulation is ineffective. However, a good-faith effort to choose and test these features appropriately should resolve this issue. One might also be concerned with user privacy. In response, we provide several comments. First, the proposed test does not require user-specific information. Second, the inputs $(\mathbf{x}, \mathbf{x}')$ need not represent real users, and we would in fact recommend that they correspond to hypothetical users. If the audit uses hypothetical users, then the main way that user information is revealed to the auditor is via the content that appears in the feeds that the auditor uses to audit because much of the content on social media is generated by users themselves. Although this issue seems unavoidable, one encouraging feature of the audit is that the auditor only requires access to the feature vectors (or embedding) of each piece of content. Therefore, as long as the auditor has no intention of unmasking the identity of users, the test could be run over these features, and this data could be immediately discarded afterwards. The only output that would be preserved is the outcome of the audit. Lastly, one implicit source of bias could be in the selection of the model family $\Theta$, which is a decision made by the auditor. We choose to leave $\Theta$ unspecified because doing so means that our analysis can be generalized to any $\Theta$ of interest. However, the auditor should test different choices of $\Theta$ and observe the outcomes. Recall that $\Theta$ captures the set of possible generative models (or, in the context of Section 4.2, possible cognitive models). In choosing the model family $\Theta$, a simple $\Theta$ is more tractable and interpretable while a complex $\Theta$ is more general.