# OpenReview forum: "Regulating algorithmic filtering on social media"
_NeurIPS.cc/2021/Conference — NeurIPS 2021 Spotlight_

### Official Review · Reviewer_AKdk · 2021-07-15

**Rating:** 6
**Confidence:** 3

**Summary:**


In this paper, the author studies the adjustment of content filtering algorithms on social platforms and proposes a system of "one platform, one user and one auditor" to study these problems in response to two problems. The author proposes a Regulatory procedure, and on this basis, studies the impact of content on users, supervision costs, and content diversity. Finally, analysis shows that content diversity plays an important role in coordinating the interests of regulatory agencies and platforms.

**Main Review:**


In general, I think the work is well-written and quite interesting - solving an emerging and important problem from a new perspective. However, both the hypothesis and the technical solution are lacking enough contributions of novelty. For example,  besides censorship and profitability, factors that affect content filtering include issues such as influence on the platform, user level, and user age. I would like to suggest the authors either make the solution actually practical or focus more on the algorithmic filtering on social networks.

**Time Spent Reviewing:**

6

---

> ### Author Response · Authors · 2021-08-11
> **In response to Reviewer AKdk**
>
> We are very thankful to Reviewer AKdk for their time and consideration. The only point we would like to make is that Reviewer AKdk's main critique in the review is addressed in our work in that the model we propose is general enough to include all the factors that Reviewer AKdk mentions. For example, as Reviewer AKdk points out, platforms may filter based on various factors, such as the influence of the user (or content creator), the user’s age, the user’s history of interaction with content on the platform, and more. These factors are certainly allowed under our model (cf. the definition of a filtering algorithm $\mathcal{F}$ in Section 3.1 is very general). As a result, our proposal and results hold when these factors are present. We thank Reviewer AKdk for their feedback, and we will work to make this point clear when revising.

---

### Official Review · Reviewer_qxt3 · 2021-07-16

**Rating:** 5
**Confidence:** 3

**Summary:**

This paper focuses on the issue of (legally) regulating filtering of content in algorithmic feeds. It assumes that feeds containing certain sets of items violate a regulation, and develops a statistical estimator which allows an external auditor to determine whether a platform complies with the regulation by comparing user feeds. Is shows theoretically that, under certain conditions, it is possible to find feeds that comply with the regulation while minimising the platform's loss (e.g. in terms of the revenue). If further shows that platforms that diversify user feeds might have an easier time complying with the regulation.


**Limitations And Societal Impact:**

As described in the main review, the paper will benefit from
* clearly articulating the modelling assumptions,
* better grounding the modeled concepts in relevant literature,
* more accurately describing its contributions in the abstract and introduction.

**Main Review:**

My relevant expertise for judging this paper lies in interpreting and computationally modelling regulations and measuring societal impacts of technologies. I am not closely familiar with statistical learning theory and would defer to other reviewers for comments on this part.

Originality:
***
A mathematical angle on social media content filtering might provide solid foundations for practical approaches, and I found the ideas developed by the paper intriguing.
***

Quality:
***
A backbone of the paper is the modelling of various social and legal phenomena, and there is room for improvement when it comes to the rigour and precision of different modelling aspects. Here are a few issues and suggestions:

1. First of all, I am not sure whether the procedure developed in the paper in practice audits what it intends to audit:
> "If an auditor is unable to detect a difference in the most gullible user’s decision making when shown the feed produced by [...], then the platform should not be penalized for the difference in those feeds."

The definitions developed early in the paper focus on what is displayed in the feed, while the auditing procedure seems to focus on the (assumed) impact a feed has on the user. These appear to be two very different concepts.


2. Currently, it is difficult to tell how certain crucial modelling choices were made: are they assumptions, are they grounded in literature outside of ML? Issues of this kind include:
- (Line 166) "Decision-making pipeline. Decisions can be reduced to a simple pipeline: (a) an individual observes information; (b) that information informs their beliefs, and (c) based on those beliefs, they make a decision."
- (Line 22) "as a user browses through their content, they implicitly learn a model of their world by regressing on the data"
It would be good to specify whether these are assumptions the paper makes or whether there exists supporting evidence in e.g. psychology literature.

3. The contributions of the paper do not accurately reflect the claims from the abstract introduction. For example, the paper does not tackle this question from the abstract at all: "In this work, we examine two questions. First, how does the filtered content affect a user’s learning and decision-making and is there a principled way to regulate this effect?". Overall, the paper would benefit by more clearly describing that the contribution lies in mathematical analysis of feed selection rather than impacts on users and their learning or decision-making.

I will let other reviewers comment on the statistical-learning-theory-related technical execution.
***

Clarity:
***
Overall, the paper is pleasant to read and well-structured. The authors clearly describe and embed their results within the story of the paper. A significant clarity issue, however, is that the paper never formally defines what a regulation is, even though it's a central concept in the paper. There are a few paragraphs that talk about the concept (lines 127 and 202), but these are either very high-level or defined through examples.
***


Significance:
***
I believe that, if the modelling quality issues described above are fixed, the paper will provide useful theoretical insights on the feasibility of feed curation as well as straightforward technical means of compliance. A few further suggestions for improving accessibility to a wider audience:
- The paper limits the analysis to regulations that can be expressed as counterfactual tests. In practice, how often do content feed regulations belong to this category?
- How many observations are needed for Algorithm 1 to confidently distinguish (non-)compliant feeds?

The result formally showing that increasing diversity might be a straightforward way to comply with content feed regulations might also have a big impact. Yet, since I don't see how exactly diversity is tied to the attribute around which regulation is defined, I am left wondering if adding diversity here is a means of complying with regulation or 'sneaking around' the regulation?


**Time Spent Reviewing:**

3

---

> ### Author Response · Authors · 2021-08-11
> **In response to Reviewer qxt3**
>
> We are very thankful to Reviewer qxt3 for their time and detailed feedback. We address a few of Reviewer qxt3’s questions below. If given the opportunity to submit a revision, we will work to make sure that these points are explained clearly in the work.
>
> 1. Reviewer qxt3’s first comment is due to a misunderstanding. Perhaps some of this confusion is clarified in the responses to 2 and 3 below. We will focus on the reviewer’s statement that “The definitions developed early in the paper focus on what is displayed in the feed, while the auditing procedure seems to focus on the (assumed) impact a feed has on the user”. The auditing procedure is based on the content that is displayed in a feed. This choice is intentional because auditing with respect to content means that the auditor requires only black-box access to the platform’s algorithm. (Requiring more than black-box access is a point of contention because platforms will generally seek to guard company information, such as its algorithms.) With respect to the notion of “(assumed) impact a feed has on the user”, we refer Reviewer qxt3 to points 2 and 3 below. The paragraphs below explain that our approach actually *removes assumptions* about users that appear in other works. This is achieved by using a helpful insight from learning theory, which says that we can focus on a concept called the minimum-variance unbiased estimator (MVUE). By focusing on the MVUE, we actually avoid making assumptions about the user, which is the basis of Section 4.1.
>
> 2. In response to Reviewer qxt3’s second point, the model we adopt is grounded in existing literature. We will work to better explain these details as well as refer to the relevant literature when revising. However, we would like to emphasize that our modeling choices are not only standard, but also that we put a great deal of effort into making our model general and minimizing the number of assumptions that we include in comparison to current works. For the purposes of this response, the comments below focus on the two lines that Reviewer qxt3 quotes.
>
>   - Line 166: This sentence is intended as a soft introduction to the mathematical formalization that follows. It simply says that, if we would like to study the effect of content on a user’s decision-making, the flowchart would look like: information --> beliefs --> decisions. As should be clear in the mathematical formalization, the proposed model is very general. For example, our model includes scenarios in which a user’s beliefs and decisions are affected by off-platform information, such as a conversation with a friend at the office or a movie that the user watched. Our model even allows for the case in which an individual is not affected by their social media content at all, e.g., perhaps the user observes the content but disregards it, or the content appears on a user’s feed but that user does not visit the social media platform on that day. This is all acceptable within our mathematical framework. In this way, Line 166 is the lightest of touches that we could conceive when studying the effect of content on a user.
>
>   - Line 22: This sentence serves to translate between intuition and statistical learning terminology. In other words, to say that content affects a user’s beliefs is to say that content affects a user’s internal model. Assuming that users have internal models is a technique used in many areas, such as microeconomics and social networks. In microeconomics, for example, a user’s internal model takes the form of a utility function, and it is often assumed to have a specific functional form (e.g., quasilinear utilities [1]). In comparison to this approach, we take a lighter touch and do not make assumptions on the precise functional form. Our approach is similar to Bayesian models of humans [2]. Yet, in comparison to Bayesian human modeling, we make even fewer assumptions in that we do not require that the human update their belief according to Bayes rule; instead, we say that the user has some update behavior $\mathcal{L}$. Another way in which statistical learning has been used to model human decision-making is in reinforcement learning, and many of these works appear in cognitive sciences (e.g., psychology) [3]. At the risk of being repetitive, the model we propose in our work makes fewer assumptions than the approaches cited above. Stated differently, the models described above are instances of our model.
>
> [1] Mas-Colell, Andreu, Michael Dennis Whinston, and Jerry R. Green. Microeconomic theory. Vol. 1. New York: Oxford university press, 1995.
> [2] Tenenbaum, Joshua B. "Bayesian modeling of human concept learning." Advances in neural information processing systems (1999): 59-68.
> [3] Steyvers, Mark, Michael D. Lee, and Eric-Jan Wagenmakers. "A Bayesian analysis of human decision-making on bandit problems." Journal of Mathematical Psychology 53.3 (2009): 168-179.
>
> 3. We disagree with Reviewer qxt3 that the first question in the abstract is not addressed in the work. How content affects a user’s learning and decision-making (the first part of the question) is the focus of Section 4.1, and we propose that this effect can be regulated (which is the second part of the question) using the well-known framework of hypothesis testing as given in Section 4.2 and analyzed in Section 4.3.  The reason that this contribution is highlighted in the abstract is that, in order to analyze how content affects users, current works make strong assumptions on how users react to their content (e.g., that a user’s opinions/actions are a weighted mean of their friends’ opinions/actions as done in DeGroot-style models [4]). Instead, we sought to adopt a more general model grounded by statistical learning theory. We found that, interestingly enough, one could bound the effects of content on *any* user by studying the MVUE. This would allow an auditor to carry out a regulatory test without (a) making strong assumptions on the way a user learns and makes decisions or (b) measuring each user’s cognitive/decision-making behaviors, which may be costly or infeasible.
>
> [4] Acemoglu, Daron, and Asuman Ozdaglar. "Opinion dynamics and learning in social networks." Dynamic Games and Applications 1.1 (2011): 3-49.

---

> > ### Comment · Reviewer_qxt3 · 2021-08-19
> > **Thank you for the rebuttal + summarized suggestions for improvments**
> >
> > Thank you for the rebuttal! In discussions, three specific suggestions for improvement have emerged, which perhaps are better summarized in a new comment:
> > - Grounding modelling choices (thank you for all the pointers above, these indeed will be very helpful to include in the paper).
> > - Adding a formal definition of what a regulation is (from my understanding above as well as reviewer iJ9f's summary, it seems to be a counterfactual over the impact of a feed on people's beliefs, but it's never formally defined).
> > - Discussing how this definition relates to how social media content is currently regulated.

---

> > > ### Author Response · Authors · 2021-08-23
> > > **Followup**
> > >
> > > Thank you! This summary (and the discussion as a whole) has been very helpful.

---

### Official Review · Reviewer_iJ9f · 2021-07-19

**Rating:** 9
**Confidence:** 5

**Summary:**

This paper studies the problem of auditing whether algorithmic filtering implementations respect a given regulation. The authors propose a simple hypothesis test that regulators could use to determine compliance given black-box access to a filtering algorithm. They show that their algorithm has desirable asymptotic properties. Finally, the authors show that under their framework platforms are incentivized to provide diverse content to their users.

**Limitations And Societal Impact:**

The authors generally did a good job of discussing the societal impacts and limitations of their work, see my main review for my one comment regarding the impacts of algorithmic filtering in the related work.

**Main Review:**

This is a very strong paper on regulating algorithmic filtering. The results are simple and elegant. The authors do a great job in explaining the intuition behind each of their results, making the paper a very pleasant read.
I especially enjoyed the framing of users as statistical estimators, the result on the minimum variance unbiased estimator (MVUE) being equivalent to the most gullible user, and the associated implications for compliance testing are likely to be extremely useful for the field moving forward. From my own personal experience, it is very hard to come up with interaction models whose assumptions are both statistically tractable and also realistic enough to be useful when considering areas that involve human interaction, the authors have done a commendable job in threading that needle.

I do have a few comments for improvement:
 - It would be nice to see some simple experiments, while I understand the paper is theoretical in nature I feel experiments would further strengthen an already strong paper, even if the experiments only involve toy data.
 - Theorem 2 is an asymptotic result, have the authors also tried providing a finite sample analysis? As it stands I don't have a good intuition for how well Algorithm 1 works for different values of m in practice, I understand that the authors have finite time but it would be good to see either experiments or a finite sample analysis (ideally both) in the final version of the paper for this reason.
 - I feel the framing of the effects of algorithmic filtering in the related work section should mention the fact that a lot of these effects are still contested. For example, this paper did not find any evidence that echo chambers were caused by recommender systems on Youtube: https://arxiv.org/pdf/2011.12843.pdf while this paper states that greater internet use does not seem to be associated with political polarization within the US https://www.pnas.org/content/114/40/10612

Finally here are some low importance nitpicks:
 - In Section 4.1, the argument for why the MVUE is equivalent to the most gullible user, and hence is the only user that needs to be considered when ascertaining whether a platform meets a regulation could be clarified. I think it's pretty clear after thinking about it for a bit, but I had to pause for a bit on my first pass through the paper.
 - Given the frequent use of footnotes to justify assumptions, I would recommend the authors simply have a separate section that discusses their assumptions. This is just a stylistic preference, so the authors can ignore this point if they wish as it did not impact my review score.

**Time Spent Reviewing:**

4

---

> ### Author Response · Authors · 2021-08-11
> **In response to Reviewer iJ9f**
>
> A sincere thank you to Reviewer iJ9f for their time and feedback. We truly appreciate the suggestions for improvement, and a brief response is provided below.
>
> We agree that experiments would bolster the work. We did in fact run several simulations that demonstrate our results on toy data, and we would be happy to include these experiments in the revision (they did not appear in the submitted manuscript due to a game-time decision to focus on the theoretical contributions).
>
> Reviewer iJ9f makes a keen observation that Theorem 2 is an asymptotic result. A finite-sample analysis would be an excellent direction to explore. In order to provide finite-sample results, we would need to restrict the parameter space $\Theta$. In other words, as one might expect, providing a more fine-grained analysis would require an additional modeling assumption. We are happy to say that we have already begun work in this direction. We have a few results on multivariate Gaussians, and we also began extending these results to exponential families (which include Gaussian and Poisson distributions, among others). We feel that it may be appropriate to reserve these results for future work in order to make sure that we do not sacrifice the clarity of the current manuscript, but we are highly encouraged by the fact that Reviewer iJ9f mentions that a finite-sample analysis would be of interest.
>
> Reviewer iJ9f’s point on related work is well taken. We are happy to receive additional references and will include the suggested works as well as highlight other areas in the space that are under debate in a revision.

---

> > ### Comment · Reviewer_iJ9f · 2021-08-12
> > **Clarifying comments**
> >
> > I am satisfied with your response to my review. I do think that toy experiments would strengthen the paper, and I look forward to seeing them included in the final version.
> >
> > Since other reviewers have indicated that it might be helpful for me to summarize Section 4 without appealing to conventions from learning theory/microeconomics I have done so below. Please correct me if I have mischaracterized your work in any way.
> >
> >
> > Problem setting:
> >
> > We're considering the setting where an auditor wants to test whether an internet platform passes regulation.
> > We have three key assumptions here:
> > 1. The auditor can craft user accounts with different properties and then query the platform for feeds this user account would observe. Note that user accounts are distinct from users here, the auditor can not query any actual users.
> > 2. The auditor has some reasonable way to represent the data generation process for each item in the feed as a probability distribution $z_i \sim \mathbb{P}(.; \theta)$ that depends on some hidden $\theta$. As a cartoon example, let's say we're interested in the average height of adults. We could have $\theta=\mu, \in \mathbb{R}$, and (assuming that each item in the feed states the height of an adult)  $z_i \sim \mathcal{N}(\mu, \sigma)$ (with $\sigma$ known).
> > 3. The regulation can be phrased in the following form: "If I vary X for a given class of users, then their belief about Y should not significantly vary based upon their feed." Here X is a property entered in the account (e.g. gender, location, interaction history with a specific type of content etc..), and Y can be any belief that estimates $\theta$ from assumption 2.
> >
> > Auditing Procedure:
> >
> > Using the cartoon example with height, the auditor might be asking whether the platform satisfies the regulation that:
> > "No matter the religion you have entered into your profile's infobox, your belief about the average height of adults should not vary significantly based upon the information you see in your feed."
> >
> >  - They would craft multiple user profiles (or they could just randomly sample existing profiles and use them as templates) that vary across a multitude of different attributes (not just religion).
> >  - Then for each user profile, they would query the platform for that user's feed, then change their religion field and query the platform for the counterfactual feeds using the modified profile.
> >  - Finally, they would run Algorithm 1 for each user which outputs whether the test is passed or not. If enough users pass the test they would certify the regulation as passed.
> >
> >
> > Key insight:
> >
> > The key insight for Section 4 is that the statement "belief about Y should not significantly vary based upon their feed" does not need to take into account all possible user learning behaviors. Instead, the authors state that we only need to consider the output of the minimum variance unbiased estimator (MVUE) on each feed. The MVUE is a function that takes as input some feed of items: $z_1, z_2, \ldots, z_m$ drawn iid from $\mathbb{P}(.; \theta)$, whose output $\hat{\theta}$ satisfies: $\mathbb{E}[\hat{\theta} - \theta] = 0$ and minimizes $\mathbb{V}[\hat{\theta}]$.
> >
> > Here's some intuition as to why we only need to consider the MVUE. Let's say a user's estimate of $\theta$ is different from the output of the MVUE, then the user's estimate can be different in two ways: it could be biased and its variance could be higher.
> >
> > If the user's estimate of $\theta$ is biased then it will consistently be biased by a fixed amount no matter the feed (this is just the definition of bias), in our height example this could consist of a user who, no matter what feed you show them would consistently over-estimate the average height by 10cm. Hence we can safely assume users are unbiased since we are only interested in how much the feed changes beliefs.
> >
> > If the user's estimate has higher variance than the MVUE, then changing their feed will probabilistically have less impact on their prediction. In our height example this could be a user who scrolls very quickly, so they only observe half of the heights in their feed.
> >
> > For a more graphical intuition, think of what would happen if I shift a 1d Gaussian's mean by 1 if it has 0 variance vs. if it has a variance of 10000. You would notice that shift in the 0 variance case with a single sample but it would take a lot of samples to notice that shift in the 10000 variance case.

---

> > > ### Author Response · Authors · 2021-08-13
> > > **Followup**
> > >
> > > Thank you for this summary and example. It is indeed accurate and captures the intuition perfectly. We've come to realize that the manuscript would benefit from a clearly stated example, as done above, and our plan is to do so in combination with toy experiments.

---

### Official Review · Reviewer_ePHk · 2021-07-23

**Rating:** 7
**Confidence:** 2

**Summary:**

In this paper, the authors study issues on regulating algorithmic filtering in social networks. They describe a system involving three actors — user, auditor, and social media platform. They prove a result claiming that a regulator only needs to test regulation for the "most gullible user"; then, for a specific class of regulations, they give an algorithm that they claim tests the regulation for said user. Finally, they give some technical conditions on when regulations do not affect the reward function of the social media platform.

**Limitations And Societal Impact:**

Yes

**Main Review:**

Strengths:

- Regulation of algorithmic filtering is a very important topic, and authors propose a seemingly novel model/have interesting technical results

Weaknesses:

- Writing is *very* confusing, to the point that the the model/problem statement/technical results are impossible to understand
- Significance of technical results seem somewhat unrelated to qualitative explanations (eg Prop 1 vs "most gullible user")

Overall, while I very much like the topic of algorithmic filtering in social networks + regulation, as well as the approach to model/analyze it using a simple 3 actor system, I find the actual technical details impossible to understand. It is not clear to me what the model even is, or what specific problem is being solved. Instead, the authors seem to go back and forth between broad definitions of regulators and social network feeds, and seemingly unrelated technical statements about hypothesis testing. While part of this is probably due to my lack of familiarity with the much of the cited literature, I do think the writing could be greatly improved.

I do sincerely think the work is interesting, so below I list some questions/comments that I hope the authors can address during the rebuttal period. I will give the paper a temporary "reject" for now, but I hope the rebuttal will make me improve my score.

I only write my comments for Sections 3 and 4, since Section 5 seems to build off the models in these sections. I also will note that I did not read the proofs carefully, although I take a look at them.

------
UPDATE: After reading the author response and the helpful summary by one of the reviewers, I have identified some misunderstandings I had in my initial review of the paper. I quite like the results in the paper now, and I have accordingly raised my score to an "accept". However, I do think the authors should provide much more context for their setting/model (eg as they have done in their responses to me and reviewer qxt3). It might also help to include an explanation like Reviewer iJ9f's for those unfamiliar with the literature (at least in the supplement).

===============

Before I get into specifics, I have to say that the fundamental issue I have with the writing is that it is never explicitly written what exact problem the auditor is trying to solve. I guess lines 223-225 attempt to, but it involves some new $f_k, f_l$ variables which don't seem connected to any of the previous definitions? (It also involves only the "most gullible user" and not all users.)

If I had to write out a specific condition for passing regulation, it would probably be something like:

>a platform passes regulation if, for each user u, their belief $\theta^{(t)}_u(Z_1, \dots, Z_t)$ at time $t$ after observing the feeds $Z_1, \dots, Z_k$ (under some user belief model) is the same as their belief $\theta^{(t)}_u(W_1, \dots, W_t)$ after observing the counterfactual feeds $W_1, \dots, W_k$.

I'm not sure how to define "counterfactual feeds" (lines 215-219 say it's out of the scope of the paper), or how the user belief function $\theta^{(t)}$ works, but this already is a cleaner statement than the hand-waving in Sections 3,4.

Now, it would be good if Prop 1/Thm 2 combined could prove a result like this. But it is not clear to me if they can. It would be disappointing if they cannot.

===============

Section 3:

— The description of what it means for whether "a platform meets regulation" is far too minimal.

>The auditor’s goal is to test whether the platform meets regulation.

This seems fine, but it's not a very specific statement. From the following "Platform" paragraph, it makes it seem like passing regulation means testing whether a feed $Z^{(t)}$ is in the list of allowed feeds $\mathcal{Z}^{(t)}$, which I don't think is the case.

Instead, a clear description of the types of regulation you consider and the exact definition of what it means to meet regulation should be defined here (eg see above attempt)

Section 4.1:

— The "user learning" section makes no sense to me, as much as I try re-reading it.

First, in line 172 you say "let \mathcal{P} ... denote the set of possible beliefs". But isn't \mathcal{P} a set of probability distributions? So is a belief a probability distribution? And $\theta$ is a parameter the user seeks to learn?

But then later on (lines 174-175) you say $\hat{\theta}^{(t)}$ is the user's belief at time $t$. So now $\theta$ is the user's belief... which they seek to learn? (Also what would a user be learning on a social media platform? I think you mean something about updating the user's beliefs? But then why is there an unknown belief $\theta$ and a learned belief $\hat{\theta}$?)

I was also confused by the learning function $\mathcal{L}$. This does not seem to depend on past beliefs (eg I would expect $\mathcal{L} : \mathcal{Z} \times \Theta \rightarrow \Theta$, or something like $\mathcal{L}(Z^t, \theta^{(t-1)}) = \theta^{(t)}$).

Because there are no citations, it's impossible to read another resource to figure out these details. The writing makes it seem as though this is an established model of user opinions, but does no citations mean that it is a novel model?

On a related note, the model does seem somewhat similar to opinion dynamics models, which I am quite familiar with, but which the paper does not cite. I think these models should be cited and compared against, since they often appear in machine learning/data mining fields, eg see references [1,2,3,...] below.

— I also don't understand the broader significance of Proposition 1.

First, why does it mention a user choosing between options? This was not mentioned in the model. One issue is that the regulatory model (which talks about counterfactuals) is not yet properly defined in the writing, so this result has no context.

But even if the regulatory model from Sec 4.2 was defined, I'd still find this result confusing. What does an option mean in the context of a news feed? What is the meaning of the "value" $v_0, v_1$ of an option (eg lines 184-186)?

I also don't really understand why Proposition 1 has anything to do with the "most gullible user"  (why does the MVUE correspond to such a user?). I also don't really understand why it means you only have to "test" said user and not every user. Isn't the user fixed in Proposition 1? (line 184 says "the user")

Also, statements like

>if the platform meets regulation for the MVUE, then it meets regulation for all other \mathcal{L}

seem vacuous since the regulatory model is not yet defined.

Section 4.2:

— One thing I found confusing is that Section 3 says the auditor only has access to the algorithm $\mathcal{F}$, but Algorithm 1 requires other things as inputs (eg MVUE feeds/counterfactual feeds). This is minor, but it is indicative of the regulatory model not being well defined.

— Where do these "conditions" $f_k, f_l$ (eg lines 206, 222) come from? I guess it makes sense the regulation $\mathcal{F}$ is a function, but it's never stated what its arguments are.

— Theorem 2 sjhould either be a standalone result, or explicitly reference Algorithm 1 (eg saying "assume Algorithm 1 was run". Using Algo 1 notation/results but not referencing it is confusing

There are also some weird circularity issues, since Algo 1 references the null/alt hypotheses defined in Theorem 2.

Also I notice the Theorem 2 statement involves \mathcal{P}, whose role in your model I do not understand (see above).

— I find the discussion of "similar outcomes" in lines 248-251 to be confusing. Isn't Theorem 2 saying something stronger — eg that the user beliefs (or whatever theta is) are the *same*, not just similar?

— Line 258:

>In practice, the auditor may wish to run this test over many users’ feeds

Very confused, because I thought we only needed to run this test once for the "most gullible user"? Why many users' feeds?

— line 265: the first reason has nothing to do with the MLE?

Nits

- In Sec 3, should state more clearly that \mathcal{Z} consists of m-tuples of pieces of content
- line 126: "denoted" → "is denoted"
- line 179: "an an" → "as an"
- line 181: "as an" → "is an"

[1] Minimizing Polarization and Disagreement in Social Networks, Musco et al WWW 2018
[2] Quantifying and minimizing risk of conflict in social networks, Chen et al KDD 2018
[3] Adversarial perturbations of opinion dynamics in networks, Gaitonde etal EC 2020
[4] Opinion Dynamics with Varying Susceptibility to Persuasion KDD 2018
[5] Analyzing the Impact of Filter Bubbles on Social Network Polarization, Chitra and Musco WSDM 2020

**Time Spent Reviewing:**

11

---

> ### Author Response · Authors · 2021-08-11
> **In response to Reviewer ePHk**
>
> We are very appreciative to Reviewer ePHk for the time and care with which they composed their review. We hope that the following address their comments and questions.
>
> **Responses for Section 3.**
>
> - Meeting regulation: To slightly contradict Reviewer ePHk, it is indeed correct that passing regulation means that a feed is in the set of allowed feeds $Z^{(t)}$. This does *not* mean that the auditor must enumerate through this (potentially infinite) list of allowed feeds in a brute-force fashion. Rather, the auditor would run the test in Section 4. We want to emphasize that this type of abstraction is standard: if the unrestricted feasible set is $\mathcal{Z}$, then the feasible set under a constraint is a subset $\mathcal{Z}^{(t)} \subset \mathcal{Z}$. That being said, we believe Reviewer ePHk is pointing to a more fundamental issue and asking for a upfront description of regulation in the manuscript. This feedback is very helpful, and we can indeed state what regulations we consider as well as state the conditions for passing regulation earlier in the work.
>
> **Responses for Section 4.1.**
>
> - Beliefs: Yes, a belief is a probability distribution parameterized by $\theta \in \Theta$, and $\mathcal{P}$ is the set of possible beliefs. The use of probability distributions to model beliefs as well as the use of “beliefs” to refer to probability distributions are standard (see discussion and references below). Note that the use of probability distributions is *very general*. For example, in opinion dynamics, a user’s belief/opinion $b$ is often given by a scalar. One could write this as $\mathcal{P} = \\{ b : b \in \mathbb{R} \\}$ or, using $\theta$, $\mathcal{P} = \\{ q(\cdot ; \theta) : \theta \in \mathbb{\Theta} \\}$, where $\Theta = \mathbb{R}$ and $q(\cdot ; \theta)$ is a Dirac delta distribution centered at $\theta$. Alternatively, suppose you wish to model beliefs as functions (e.g., in microeconomics, beliefs are often modeled as utility functions). A function $f(\cdot)$ *is* a probability distribution that, given x, place zero probability on all values except $f(x)$. In this way, the representation using probability distributions is flexible.
>
> - The notation and terminology that we use is fairly standard in statistical inference and learning. The term “beliefs” is often used to refer probability distributions, as popularized by Bayesian inference [1]. The use of notation such as that $\mathcal{P} = \\{ p_{\mathbf{z}}(\cdot ; \theta) : \theta \in \Theta \\}$ is standard in statistical inference and information theory, and we refer the reader to [2] (e.g., at the start of Section 2.9). One often uses the shorthand $\theta$ to refer to a belief rather than $p_{\mathbf{z}}(\cdot ; \theta)$. We will work to make sure that the manuscript points to relevant resources and thank Reviewer ePHk for bringing this to our attention.
>
>   - [1] Bernardo, José M., and Adrian FM Smith. Bayesian theory. Vol. 405. John Wiley & Sons, 2009.
>   - [2] Cover, Thomas M., and Thomas, Joy A.. Elements of Information Theory. Germany, Wiley, 2012.
>
> - Users: We model the user’s beliefs as probability distributions, and we refer to the updating of their beliefs based on information that they see as “learning”. This does not mean that the user is actively seeking to learn some unknown belief. The model simply says that a user’s belief/internal model can be represented as a probability distribution and the process of updating it is what we mean by “implicit learning”. We also wanted to thank Reviewer ePHk for the references on opinion dynamics and to mention that we include several references on the closely related field of social learning.
>
> - Depending on past beliefs: Please see footnote 4, which says that the update operation $\mathcal{L}$ can depend on previous beliefs with no change to our results.
>
> - Proposition 1: First, this decision-making does *not* have to occur on the platform. For example, the platform could show information on COVID-19, but the decision could be whether to get the vaccine. Second, the use of value functions is very general. In microeconomics, the value function would be an individual's utility function. In reinforcement learning, the value function would be the upper-confidence bound. Moreover, binary hypothesis tests are well accepted in decision theory because any decision between a finite number of options can be represented as a series of binary decisions.
>
> - MVUE: In order to analyze how content affects users, current works make strong assumptions on how users react to their content (e.g., that a user’s opinions/actions are a weighted mean of their friends’ opinions/actions [4]). Instead, we sought a more general model so that users could have any learning/belief updating behavior $\mathcal{L}$. However, there is a reason why current works have not been more general---because it is difficult to characterize the effects of content on users without these assumptions. Proposition 1 leverages a fascinating result from learning theory, which says that we can move past this obstacle by measuring how content affects a hypothetical user whose updating behavior $\mathcal{L}$ corresponds to the MVUE. The major proof insight in this result is given by [3].
>
>   - [3] Ghobadzadeh, Ali, Sayed Jalal Zahabi, and Ali A. Tadaion. "The role of MVU estimator and CRB in binary composite hypothesis test." 2009 IEEE International Symposium on Information Theory. IEEE, 2009.
>
> - In the spirit of providing more references, the following discusses decision theory and hypothesis testing.
>
>   - [4] Parmigiani, Giovanni, and Lurdes Inoue. Decision theory: Principles and approaches. Vol. 812. John Wiley & Sons, 2009.
>
> **Responses for Section 4.2.**
>
> - Start of Section 4.2: We believe this comment is due to a misunderstanding. When it comes to regulations, there is often concern that, in order to test the regulation, one may require deep access into company information (e.g., its algorithms), and such access is generally difficult to come by. Our statement that the auditor only requires black-box access to the algorithm is to emphasize that the proposed test does not require deep access. But, in order to run a black-box algorithm, one does need to give it inputs. We clearly state that our contribution is not to prescribe a regulation but to provide a translation from a regulation to a test. Therefore, the inputs depend on the regulation. Drawing an analogy to regulations on car emissions, one must give it inputs/parameters such as: (distance, speed, incline). The auditor would drive the car for that distance at the given speed on a surface with the specified incline and test whether its emissions fall within an acceptable range. The black box is the car (the auditor does not need to know how it drives), and the inputs depend on the regulation (perhaps a different regulation also cares about number of passengers).
>
> - Lines 206 and 222: The filtering algorithm is given the label $\mathcal{F}$. If we understand correctly, Reviewer ePHk is suggesting providing an explicit functional form for $\mathcal{F}$. We could write something to the effect of $\mathcal{F} : \mathcal{Y} \rightarrow \mathcal{Z}$, where $\mathbf{f}_j , \mathbf{f}_k \in \mathcal{Y}$. Let us know if this improves clarity.
>
> - Theorem 2 and circularity: Thanks for these notes. There is no circularity. Algorithm 1 introduces the notation that $\hat{H} = H_0$ if the test is passed and $\hat{H} = H_1$ if not. Theorem 2 uses similar notation to prove that the result of the audit satisfies a strong guarantee. If one would like, one could instead replace all the instances of $H$ in Theorem 2 with $F$ or $G$--- that is to say, the mirroring of notation is a style choice that we can definitely reconsider in a revision. However, we do believe that such mirroring is fairly standard.
>
> - Line 248-251: Reviewer ePHk is on the right track. A regulation that requires identical feeds across users would be far too harsh, if not impossible to enforce, because users have different content sources. Without identical feeds, it is not possible to enforce that the effects on the user are identical, but one can instead enforce that they are sufficiently similar. (If it is helpful, one can draw an analogy to differential privacy, which enforces privacy by requiring *not* that two outcomes are identical, but that they are similar when two datasets differ by only one element.) In the language of Theorem 2, the hypothesis test does not require that the null hypothesis $H_{0,j}^{(t)}$ is true but, rather, that it cannot be rejected in favor of the alternate hypothesis.
>
> - Line 258: An auditor would like to test compliance from feeds. To do so, the auditor may wish to look at many feeds that the platform (i.e., its algorithm) produces. This is what we mean by looking at many feeds. As an analogy, an auditor checking a car manufacturer’s carbon emissions may wish to test many of its cars, not just one. We intentionally design the procedure to be feasible given only one feed in order to have a procedure that is modular (i.e., can be scaled up). Explaining that the algorithm is modular and can be scaled up is the purpose of Line 258. When given a feed or several feeds, the auditor then carries out the procedure using the MVUE. We refer to the MVUE as the most gullible user, but you may wish to simply think of it as a technical concept if the use of “most gullible user” is confusing.
>
> - Line 265: If we exclude the first reason, the second reason would not be enough to standalone. The second reason states that the MLE and MVUE behave similarly asymptotically, when the MVUE exists. However, this alone would not be enough of a reason to replace the MVUE with the MLE. In order to add more rigor, it should be combined with the first reason, which states that the MVUE-MLE asymptotic equivalence is useful because the guarantee provided in Theorem 2 is proven based on the MVUE’s asymptotic behavior.

---

> > ### Comment · Reviewer_ePHk · 2021-08-12
> > **Response to author comments**
> >
> > Thanks for your detailed rebuttal to my comments, and especially for the numerous references.
> >
> > After reading your rebuttal and Reviewer iJ9f's very helpful comment, I realize I had some big misconceptions when initially reading the paper. In particular, IJ9f's running example about the belief $\theta$ being the average height helped me understand the setting in Section 4.1 much more clearly. I did not realize it was standard terminology (sorry for the initial misunderstanding). That said, I agree with the authors about adding references to Section 4.1 (eg noting the differences between other existing models, as in the response to Reviewer qxt3)
> >
> > I also understand the significance of Proposition 1 and the MVUE much better now, and I agree with the authors that Proposition 1 is a fascinating result (especially cool that it leverages a result from 10 years ago). In particular, I initially thought that Proposition 1 said that a regulator would only have to check whether only one single user passes regulation, but now I realize that it refers to a regulator needing to check only one single learning behavior $\mathcal{L}$ for a \emph{fixed} user feed. While the result is quite nice, I think the ``most gullible user" terminology (lines 191-197) contributed to my initial confusion, so it might help readability of your paper if you use different terminology.
> >
> > In any case, I will revise my rating to an "accept" now, since now that I understand the setting I quite like the results. I do agree with other reviewer comments on: (1) including experiments if space/time permits, (2) finite sample analyses (at least mentioning this as a future direction in the conclusion if there is space), (3) adding more context for the setting/modeling choices.
> >
> > =============
> >
> > Other comments:
> >
> > > Reviewer ePHk is suggesting providing an explicit functional form for $\mathcal{F}$. ... Let us know if this improves clarity.
> >
> > Yes that would be helpful.
> >
> > > Theorem 2 and circularity:
> >
> > I suppose my issue is that Algorithm 1 defines its output as $H_0$  and $H_1$ without referencing the specific hypothesis test. So if one were to read Algorithm 1 in isolation, they would not know what the null/alternative hypotheses are. My suggestion is, in Algo 1, it might be helpful to explicitly reference the hypothesis tests from Thm 2.
> >
> > > Line 248-251:
> >
> > Thanks, this is a helpful response!
> >
> > > Line 258: An auditor would like to test compliance from feeds. To do so, the auditor may wish to look at many feeds that the platform (i.e., its algorithm) produces. This is what we mean by looking at many feeds. As an analogy, an auditor checking a car manufacturer’s carbon emissions may wish to test many of its cars, not just one. We intentionally design the procedure to be feasible given only one feed in order to have a procedure that is modular (i.e., can be scaled up). Explaining that the algorithm is modular and can be scaled up is the purpose of Line 258. When given a feed or several feeds, the auditor then carries out the procedure using the MVUE. We refer to the MVUE as the most gullible user, but you may wish to simply think of it as a technical concept if the use of “most gullible user” is confusing.
> >
> > I discussed this above already, but I guess one area of confusion I had with the "most gullible user" terminology is that I would associate different feeds with different users. So the "most gullible user" terminology made me think the auditor only had to check a *single* feed.
> >
> > > Line 265:
> >
> > Thanks, this is helpful. I guess when I read this I thought the two reasons were standalone, rather than  being two parts of the same reason.

---

> > > ### Author Response · Authors · 2021-08-13
> > > **Followup**
> > >
> > > Thank you for the follow-up. This discussion has been extremely helpful in understanding which parts of the manuscript are clear and which parts need clarification, and we'll work to incorporate the suggestions above.

---

### Decision · Program_Chairs · 2021-09-27

**Decision:**

Accept (Spotlight)

**Comment:**

This paper studies the problem of auditing whether algorithmic filtering implementations respect a given regulation from a theoretical perspective. The authors propose a simple hypothesis test that regulators could use to determine compliance given black-box access to a filtering algorithm. They show that their algorithm has desirable asymptotic properties. Finally, the authors show that under their framework platforms are incentivized to provide diverse content to their users. The topic is very timely and the presented theory/methodology is a strong contribution.

Initially, some of the reviewers had concerns regarding the presentation of the results. The authors' reply as well as especially the follow-up explanation/description by one of the reviewers, who enthusiastically championed the paper, cleared up these concerns to some extent. However, it is very important that the authors incorporate the reviewers' suggestions in the final version of the paper, including the running example provided by one of the reviewers, so that a wider audience can appreciate the contribution.